# Biofilms as Promoters of Bacterial Antibiotic Resistance and Tolerance

**DOI:** 10.3390/antibiotics10010003

**Published:** 2020-12-23

**Authors:** Cristina Uruén, Gema Chopo-Escuin, Jan Tommassen, Raúl C. Mainar-Jaime, Jesús Arenas

**Affiliations:** 1Unit of Microbiology and Immunology, Faculty of Veterinary, University of Zaragoza, Miguel Servet, 177, 50017 Zaragoza, Spain; 700469@unizar.es (C.U.); 721072@unizar.es (G.C.-E.); rcmainar@unizar.es (R.C.M.-J.); 2Department of Molecular Microbiology, Institute of Biomembranes, Utrecht University, 3584 CH Utrecht, The Netherlands; J.P.M.Tommassen@uu.nl

**Keywords:** biofilms, antibiotic resistance, antibiotic tolerance, multidrug-resistant bacteria, recalcitrance, biofilm control

## Abstract

Multidrug resistant bacteria are a global threat for human and animal health. However, they are only part of the problem of antibiotic failure. Another bacterial strategy that contributes to their capacity to withstand antimicrobials is the formation of biofilms. Biofilms are associations of microorganisms embedded a self-produced extracellular matrix. They create particular environments that confer bacterial tolerance and resistance to antibiotics by different mechanisms that depend upon factors such as biofilm composition, architecture, the stage of biofilm development, and growth conditions. The biofilm structure hinders the penetration of antibiotics and may prevent the accumulation of bactericidal concentrations throughout the entire biofilm. In addition, gradients of dispersion of nutrients and oxygen within the biofilm generate different metabolic states of individual cells and favor the development of antibiotic tolerance and bacterial persistence. Furthermore, antimicrobial resistance may develop within biofilms through a variety of mechanisms. The expression of efflux pumps may be induced in various parts of the biofilm and the mutation frequency is induced, while the presence of extracellular DNA and the close contact between cells favor horizontal gene transfer. A deep understanding of the mechanisms by which biofilms cause tolerance/resistance to antibiotics helps to develop novel strategies to fight these infections.

## 1. Introduction

During the last few decades, a significant increase in the number of clinical and environmental multidrug-resistant (MDR) bacteria, also called superbugs, has been reported. Particularly problematic in this respect are the major human pathogens, e.g., *Enterococcus faecium*, *Staphylococcus aureus*, *Klebsiella pneumoniae*, *Acinetobacter baumannii*, *Pseudomonas aeruginosa*, and *Enterobacter* spp. Additionally, among veterinary pathogens, particularly those associated with livestock farming and poultry production, the rate of drug resistance has increased. Some of them, such as *Salmonella enterica* and *Campylobacter* spp., are also important zoonotic pathogens. 

About 700,000 human deaths are attributed to MDR bacteria each year globally, and this number is expected to exceed 10 million deaths by 2050, at a cumulative global cost of USD 100 trillion [1]. Additionally, the Global Antimicrobial Surveillance System from the World Health Organization (WHO) reported a widespread occurrence of MDR in 2018 among 2,164,568 people with suspected bacterial infections across 66 countries (ranging from high to low income). The proportion of people infected with MDR bacteria varies among countries, with higher levels of resistance to those drugs more widely utilized for treating infections (e.g., ciprofloxacin) [2], and many pathogens have developed mechanisms to survive to practically all of the antibiotic families available on the market, for example *K. pneumoniae*. Despite the apparent exacerbation of this global issue, the high costs of the development of new antibiotics and the unavoidable emergence of antimicrobial resistances (AMR) have caused antibiotic development to lack economic appeal to the pharmaceutical industry. 

MDR infections have been extensively associated with hospital and healthcare settings, for example those caused by *P. aeruginosa*, *A. baumannii*, or *K. pneumoniae*. However, MDR pathogens can also be transmitted within the community, for example *Neisseria gonorrhoeae.* In addition, people can be infected by MDR zoonotic bacteria that were selected through the use of antibiotics in food animals. In fact, antimicrobial resistance (AMR) in zoonotic bacteria recovered either from food animals or their carcasses is widespread. For instance, among *Salmonella enterica* recovered from pig carcasses in the EU in 2017, 53%, 59.5%, and 56.8% were resistant to ampicillin, sulfamethoxazole, and tetracycline, respectively. In the case of *Campylobacter* spp. recovered from poultry meat, high levels of resistance were noted for ciprofloxacin, nalidixic acid, and tetracycline in 54–83% of the isolates [3]. In fact, the relation between the use of some antibiotics in food animals and the subsequent detection of AMR in bacteria isolated from humans has been suggested in several studies [4,5]. Thus, the wide spreading of AMR, the high mortality rates, and the lack of initiatives to discover new antibiotics together make MDR bacteria a critical problem for modern medicine.

In addition to the well-known genetic mechanisms behind the AMR phenomenon and the transfer of antibiotic-resistance genes by horizontal gene transfer (HGT), bacteria are capable of displaying other strategies to withstand an exposure to antimicrobials, one of which is the ability to produce biofilms. Biofilms are highly structured associations of microorganisms embedded in a self-produced extracellular matrix (ECM) and adhered to a biotic or an abiotic surface. The biofilm confers many benefits to the members of the community, including collective recalcitrance. Recalcitrance is a term defined as “*the ability of pathogenic biofilms to survive in presence of high concentrations of antibiotics*” [6]. Indeed, biofilm cells are 10–1000 fold less susceptible to various antimicrobial agents than their planktonic forms [7,8,9]. Recalcitrance to antibiotics is achieved in biofilms through a variety of mechanisms, some of which can lead to an increase in the number of MDR bacteria, since processes such as HGT or hypermutability are favored within the biofilm environment. In fact, the biofilm is recognized as a reservoir of antibiotic-resistance genes [10].

Biofilms are involved in a broad range of infections. Indeed, about 80% of the chronic and recurrent microbial infections in humans are caused by biofilms, some of which result in high mortality and morbidity rates [11,12], particularly those caused by MDR bacteria. Patients with cystic fibrosis or with assisted ventilation are susceptible to chronic infections by biofilm-forming *P. aeruginosa* [13] and *A. baumannii* [14], respectively. These bacteria are also well-known for acquiring MDR, and the resulting biofilms are almost impossible to treat [15]. In addition, biofilms are highly resistant to the host immune defenses and clearance mechanisms. Other nocosomial infections include those caused by biofilms strongly adhered to implants and catheters or medical devices. They are generated by different pathogens, such as *Escherichia coli*, *Proteus mirabilis*, or *K. pneumoniae* [16,17], which can also become MDR. Additionally, *N. gonorrhoeae* forms biofilms on genital mucosa causing chronic infections [18]. The WHO classified this pathogen as high priority because of its extraordinary capacity to persist against all recommended antibiotics to treat the infection [19]. Together, these and many other studies have related the high capacity of several bacteria to survive to antibiotics through biofilms. Understanding the mechanisms that these pathogens utilize to survive antibiotics will help to design adequate surveillance methods and novel strategies to combat these infections. This review focuses on the role of biofilms in the poor susceptibility of bacteria to antibiotics. We will describe the mechanisms responsible for recalcitrance and the currently proposed solutions to treat or prevent biofilm infections. 

## 2. Basis of Biofilm-Mediated Antibiotic Survival

The recalcitrant nature of biofilms to antibiotics depends mostly on (i) the developmental stage of the biofilm, (ii) the ECM composition, and (iii) the biofilm architecture. 

### 2.1. Biogenesis of Biofilms 

Biofilm formation is a dynamic process that takes place in a series of sequential steps. It is initiated by the interaction of the bacteria with a surface. Exposure of planktonic cells to stress, which may be provoked by antibiotics, starvation, or other adverse environmental conditions, can initiate biofilm formation by activating gene expression [20]. Additionally, molecules involved in cell-to-cell communication accumulate at high cell density. These molecules, generally referred to as autoinducers, can activate and regulate the process [21] and allow for a coordinated response of the population members, which is known as quorum sensing (QS). The first step of biofilm formation consists of the adhesion of the bacteria to the substratum. This process is often mediated by long, proteinaceous, filamentous fibers that protrude from the bacterial cell surface, such as flagella, fimbriae, or pili. After initial interaction is established, shorter cell surface-exposed structures interact with the substratum, thereby increasing the contact between bacteria and the substratum [12]. Strains of *E. coli* and *Salmonella* produce curli fimbrae that mediate both cell-to-substratum and cell-to-cell interactions [22]. Other proteins such as Bap-family proteins in *S. epidermidis* [23] or CdrA in *P. aeruginosa* [24] are large proteins that interact with ECM components and the bacterial cell surface thereby strengthening the matrix. Autotransporters are proteins secreted through the Type V secretion system in many Gram-negative bacteria and often have demonstrated roles in interbacterial interactions [25]. Then, the bacteria secrete ECM components and proliferate to form a microcolony. ECM serves as a glue element that helps to stabilize interbacterial interactions. Bacteria within the microcolony communicate and organize spatially. Type IV pili act at the junction between cells to form microcolonies and can also contribute to the reorganization of bacteria within the biofilm [26]. Cell-to-cell communication, including QS [21] and also cell-contact-dependent communication systems [27], seem to be relevant for this process. At this stage, the expression of genes for the formation of the ECM increases [12], and biofilms become less vulnerable to antibiotics than earlier biofilm stages [28].

### 2.2. Composition of the ECM 

The ECM consists of a conglomerate of different substances that together provide structural integrity to the biofilm. In general, the ECM can be composed of water, polysaccharides, proteins, lipids, surfactants, glycolipids, extracellular DNA (eDNA), extracellular RNA, membrane vesicles, and ions such as Ca^2+^. In many bacteria, extracellular polysaccharides and eDNA are prominent components of the ECM [29].

eDNA is constituted of chromosomal DNA that is released into the extracellular milieu through cell lysis, dedicated secretion systems, or membrane vesicles. eDNA is often involved in adhesion, particularly after the first interaction of the cell with the substratum. It mediates acid–base interactions and increases the hydrophobicity of bacterial cells which are favorable for the cell–substratum interaction [30,31]. Indeed, eDNA is used for initiation of biofilm formation in many pathogenic bacteria, including Gram-positive and Gram-negative bacteria and mycobacteria [32,33,34]. In addition, eDNA facilitates the interaction of the bacteria in the ECM. This is achieved by binding of positively charged segments of cell surface-exposed proteins with the negatively charged eDNA molecules [35]. Various proteins can be implicated in this interaction, such as autotransporters, lipoproteins or two-partner secretion protein A of Gram-negative bacteria, and cell wall-associated proteins in Gram-positive bacteria and fungi [35]. Thus, anchoring the eDNA to the cell surface by DNA-binding proteins is a widespread mechanism for biofilm formation that may also facilitate multispecies biofilms. eDNA can also mediate interactions with other ECM components such as polysaccharides [36]. Together, these interactions are relevant for the structural integrity of biofilms.

The composition of the polysaccharides present in the ECM varies between different bacterial species and even between different isolates of the same species. Most are long linear or branched molecules formed by one (homopolysaccharides) or several different (heteropolysaccharides) residues. They may contain substituents that greatly affect their biological properties. One of the most commonly studied polysaccharides is poly-β-1,6-N-acetyl-D-glucosamine, often named PGA or PNAG. It is synthetized by *E. coli* [37] and *S. aureus* [38], among others. In *E. coli*, PGA is required for initial cell-to-cell and cell-to-substratum attachment [37]. Another polysaccharide present in ECM is cellulose, a linear polymer of β-1,4 linked D-glucose. It is a major component of the ECM of some *E. coli* [39], *Salmonella* [40], and *Pseudomonas* strains [41]. Some *E. coli* strains produce a complex branched polysaccharide called colanic acid [42]. Additionally, *P. aeruginosa* can produce diverse exopolysaccharides. Mucoid *P. aeruginosa* strains produce alginate, a polymer of β-1-4-linked mannuronic acid and α-L-guluronate. Production of alginate confers a mucoid phenotype [43,44], typical of strains isolated from lungs of cystic fibrosis patients with *Pseudomonas* infections that underwent several rounds of antibiotic treatment. Therefore, secretion of alginate is related to pathogenic biofilms [45]. Alginate mediates the establishment of microcolonies at early stages of biofilm formation and provides stability to mature biofilms. Nonmucoid *P. aeruginosa* strains can produce other exopolysaccharides, e.g., Psl or Pel. Pel is a linear, cationic exopolysaccharide formed by 1→4 glycosidic linkages of *N*-acectylglucosamine and *N*-acetylgalactosamine. It has a critical role in maintaining cell-to-cell interactions and pellicle formation [46]. In contrast, Psl is composed of repeating pentasaccharide subunits of D-glucose, D-mannose, and L-rhamnose [47]. Psl mediates attachment to biotic surfaces such as mucin-coated epithelial surfaces and epithelial cells, indicating its relevance for the establishment of *P. aeruginosa* infection [48]. Additionally, *P. aeruginosa* strains can secrete cyclic and linear glucans [49,50] that are formed by β-1,3 linked glucose residues.

The proteinaceous content of the ECM includes proteins that are secreted through active secretion systems or released during cell lysis. The role of many of these proteins in the biofilm matrix is unknown, but some of them have been identified as important contributors to biofilm formation or restructuring in many pathogens. Various are extracellular enzymes. Their substrates can be polysaccharides, proteins, and nucleic acids, present in the ECM. They can function in remodeling of the ECM, detachment of cells from the biofilm, or degradation of polymers for nutrient acquisition. 

ECM biogenesis and composition are dynamic and vary between strains of a given species and also depend on environmental conditions, such as nutrient availability and the presence of stressors, and on social crosstalk. Several functions have been attributed to the ECM based on its extraordinary capacity to establish intermolecular interactions between its components, and with surface-exposed structures of the cells, biotic and abiotic substrata, and many environmental molecules [29]. Thereby, the ECM immobilizes cells and keeps them in the biofilm community. By retaining the cells in close proximity, the ECM establishes the optimal conditions for interbacterial communication and exchange of genetic material, which is relevant, amongst others, for the dispersion of antibiotic-resistance genes. The ECM additionally retains water and thereby protects the cells against desiccation. Furthermore, the extracellular enzymes in that hydrated environment generate an external digestive system. In addition, ECM retains several other substances, for instance, nutrients, energy sources, antibiotics, antibiotic-degrading enzymes, and molecules released by cell lysis, thereby constituting a recycling unit [29]. In general, the ECM acts as a protective scaffold.

### 2.3. Biofilm Architecture 

The architecture of biofilms is defined by the organization of the biomass and the spaces in between. The development of this structure depends on the composition of cell-surface structures mediating mutual interactions between cells and interactions of cells with ECM components and with the substratum [35]. The biofilm architecture is responsible for the generation of gradients of dispersion of substances within the biofilm. This will influence the accessibility of these substances to particular niches inside the biofilm, and determines, amongst others, the variation in antibiotic susceptibility of cells within biofilms. Figure 1 illustrates the biofilm architecture of different bacterial species. *P. aeruginosa* strain ATCC 15,692 forms complex biofilms with mushroom-like architectural features consisting of well-defined stalks and caps. *Enterococcus faecalis* ATCC 51,299 biofilms, however, are flat and compact [51], while *Salmonella enterica* strain S12 and *E. coli* strain ESC.1.16 form biofilms constituted of small cell clusters (Figure 1A) [51]. In contrast, biofilms of *Neisseria meningitidis* strain HB-1 are constituted of cell aggregates of different sizes forming defined channel-like structures [33] (Figure 1B). 

Thus, in general, based on their architecture, biofilms can be classified into (i) monolayer biofilms, formed by a compact layer with high surface coverage, or (ii) multilayer biofilms, formed by bacterial clusters of different morphology with a low surface interaction. The biofilm architecture can vary depending on different factors, for instance the expression of surface-exposed proteins. Examples are the meningococcal autotransporters AutA and AutB, whose expression is phase variable and significantly alters the biofilm (Figure 1B) [52,53]. Additionally, the medium composition influences the biofilm architecture. *P. aeruginosa* PAO1 makes monolayer biofilms in the presence of citrate benzoate and casamino acids and multilayer biofilms in presence of glucose [54].

## 3. Mechanisms of Biofilm Recalcitrance

### 3.1. Types of Antibiotic Recalcitrance 

Biofilm recalcitrance comprises two independent phenomena: antibiotic resistance and antibiotic tolerance. Resistance refers to the capacity of a microorganism to survive and grow at increased antibiotic concentrations for long periods of time and is quantifiable by assessing the minimum inhibitory concentration (MIC) [55]. It involves mechanisms that prevent the binding of an antibiotic to its target, including enzymatic deactivation, active efflux of a drug once it is in the cytoplasm or the cytoplasmic membrane, or reduced influx, among others, and can be generated by HGT or mutations. Together, they preclude antibiotics from altering their target’s function and they prevent the production of toxic products that would end up damaging the cell. Resistance can be further classified into intrinsic, acquired, and/or adaptive resistance (expanded in Box 1). 

Box 1Types of antibiotic resistance.**Intrinsic resistance.** This is the inherent/natural property of the bacteria to withstand antibiotics. For example, Gram-negative bacteria are, in general, more resistant to antibiotics than Gram-positive ones due to the presence of the outer membrane, which reduces the permeability to many antibiotics [56]. Another example is the wall-less bacterial genus Mycoplasma, which is not affected by antibiotics whose target is the cell wall.**Acquired resistance.** This arises through genetic modifications of originally sensitive bacteria, either through mutations or by the incorporation of new genes via HGT. Thus, microorganisms that are initially sensitive to an antibiotic become resistant due to spontaneous or induced mutations that alter, for example, the target of the antibiotic or its uptake by the cell or after the acquisition of one or more molecular mechanisms for AMR, such as antibiotic inactivation or, increased antibiotic efflux [57]. These genetic modifications are heritable and will result in a permanent effect if the fitness-cost associated with them is low or null or compensation mechanisms exists [58].**Adaptive resistance.** This is the capacity of bacteria to vary rapidly gene expression or protein production in response to antibiotics or adverse environmental conditions. The molecular mechanism behind involves epigenetic inheritance, population heterogeneity, gene amplification, and efflux pumps that are regulated by intricate regulatory pathways [59].**Heteroresistance.** This is the presence, within a given population of bacteria, of one or several subpopulations displaying increased levels of antibiotic resistance compared with the main population [60]. This phenomenon is often related with the presence of unstable genes which would give the bacteria a high likelihood for reversion to susceptibility in the absence of antibiotic selective pressure [61]. This instability makes its detection difficult, increasing thus the risk for treatment failure [62].

By contrast, antibiotic tolerance is the capacity of bacteria to survive a transient exposure to increased antibiotic concentrations, even those above the MIC. Tolerance is assessed by the minimum bactericidal concentration, that is, the minimum concentration of antibiotic required to kill 99.9% of the cells [63]. Unlike resistance, tolerance is only temporary and after longer exposure periods, the antibiotic will kill the bacteria. It is an adaptive phenomenon that implies a change in cellular behavior, from an active (growing) state to a quiescent (dormant) state [57], and requires large metabolic rearrangements affecting, for example, energy production and nonessential functions. These changes are triggered during poor growth conditions or exposure to stress factors or antibiotics [55,64]. In this case, antibiotics can usually attach to the target molecules, but because their function is no longer essential, the microorganism survives [65]. Tolerance in biofilms is also caused by entrapment of the antibiotics in the ECM, in this case, the antibiotic does not reach its target. In contrast to resistant cells, tolerant cells within the biofilm cannot grow in presence of a bactericidal antibiotic. Persistence is an especial phenomenon of tolerance [64]. Indeed, persistence is a phenomenom that increase the survival of a given population in the presence of bactericidal antibiotics without enhancing the MIC, but in contrast to tolerance, persistence only affects a subset of cells of the population called persisters [64]. Persisters cells are tolerant cells that eventually can be killed at longer exposure times. There are two types of persisters, e.g., type I or triggered persistence, which is induced upon environmental signals, such as starvation, and type II or spontaneous persistence, where a subpopulation of growing bacteria converts into the persister state by a stochastic process [66]. Any how, persistence can be also refereed to as heterotolerance, which is different than heteroresistance (Box 1), as persisters can eventually be killed at longer exposure times. Figure 2 illustrates the mechanisms that govern antibiotic tolerance and antibiotic resistance of biofilms, and they will be further discussed in the next section. 

Overall, biofilm recalcitrance does not depend on one unique mechanism but is a combination of both antibiotic tolerance and antibiotic resistance mechanisms. Such combination varies depending upon aspects such as the bacterial species or strain, the antimicrobial agent, the developmental stage of the biofilm, and the biofilm growth conditions [63]. 

### 3.2. The Protective ECM Barrier

The components of the ECM can impact the efficacy of antibiotics on biofilm-forming cells. ECM influences the biofilm architecture, which in turn generates gradients of dispersion that affect the access of antibiotics to the biofilm members. The flow of substances through biofilm varies. In the channels, the flow is convective while it occurs by diffusion within cell clusters [67]. Thus, antibiotics can penetrate rapidly through the channels of the biofilm but may be retained locally in cell aggregates. Due to the differences in biofilm architecture (Figure 1), the time for an antibiotic to reach the interior of the biofilm varies between strains [68]. 

The diffusion of antibiotics through the biofilm can also be limited by their interaction with particular ECM components, which affects antibiotic effectivity. This is illustrated in the literature by several examples including *P. aeruginosa.* Alginate is a polyanionic exopolysaccharide that protects *Pseudomonas* biofilms from aminoglycosides [44]. Additionally, the highly anionic cyclic glucans present in these biofilms interact with the aminoglycoside kanamycin [69]. Presumably, the high negative charge of alginate and the cyclic glucans helps to establish ionic interactions with these positively charged antibiotics. However, in strains that do not secrete alginate, the polysaccharides Pel and Psl are involved in the establishment of biofilms. Pel provides protection against the aminoglycosides tobramycin and gentamicin, but not against ciprofloxacin [70]. Unlike alginate, Pel is a cationic exopolysaccharide, and, therefore, resistance to aminoglycosides cannot be explained by direct charge interaction. However, Pel binds eDNA [46], which is negatively charged and can interact with aminoglycosides. Additionally, Pel could bind negatively charged portions of other antibiotics such as ampicillin, though this hypothesis has not been explored yet. The exopolysaccharide Psl also binds eDNA [71] and plays a role in resistance to colistin, polymyxin B, tobramycin, and ciprofloxacin at early stages of biofilm formation [72]. This protective effect was also observed in the non-Psl producing species *E. coli* and *S. aureus* if they were present in a mixed biofilm together with *P. aeruginosa* [72].

Studies in *Pseudomonas* evidence that eDNA enhances resistance of biofilms to aminoglycosides but not to fluoroquinolones or β-lactams [73,74]. eDNA also enhances resistance of biofilms of *Staphylococcus epidermidis* to the glycopeptide antibiotic vancomycin [75]. Likely, the negatively charged eDNA binds positively charged aminoglycosides and glycopeptides. The latter study [75] also demonstrated that the binding constant of vancomycin and eDNA is up to 100-fold higher than that of vancomycin and its target, the D-Ala-D-Ala peptide in peptidoglycan precursors. Thus, within the biofilm environment, eDNA may compete with D-Ala-D-Ala peptide, and, because of the higher affinity of vancomycin for eDNA, it may be retained in the ECM. Besides direct interaction of eDNA with antibiotics, accumulation of eDNA creates a cation-limited environment by chelating cations such as Mg^2+^. In *P. aeruginosa* and *S. enterica* serovar Typhimurium (*S.* Typhimurium), reduction of the Mg^2+^ concentration triggers the two-component regulatory systems PhoPQ and PmrAB, which are linked to AMR [74,76]. Activation of these systems generates modifications in the lipid A moiety of lipopolysaccharides (LPS) through (i) the expression of the outer membrane (OM) protein PagP, which adds a palmitoyl residue to the lipid A, (ii) the substitution of the phosphate groups with 4-amino-4-deoxy-L-arabinose and/or phophoethanolamine, and (iii) the production of LpxO, which adds a hydroxyl group onto the second carbon atom of one of the fatty acyl chains. The first stage of aminoglycoside uptake involves the binding of the polycationic antibiotics to the negatively charged components of the bacterial membrane, such as LPS of Gram-negative organisms. This is followed by displacement of Mg^2+^ ions [77,78], which leads to disruption of the OM and initiation of aminoglycoside uptake [79,80]. The modifications in the lipid A generated by the activation of PhoPQ and PmrAB alter considerably the lipid A charge and the OM permeability, which could explain the involvement of eDNA in aminoglycoside resistance. On the other hand, the activity of antimicrobials can also promote the release of eDNA to the ECM. For instance, the amount of eDNA in biofilms of S. epidermidis doubled by treatment with vancomycin [75]. The released eDNA can then bind the positively charged antibiotics and prevent them from reaching the cells and exerting their activity. Considering that eDNA is a constituent of the ECM of many bacteria, it is tempting to speculate that this phenomenon could also occur in other bacteria. Additionally, antibiotic-modifying enzymes can be released and located into the biofilm matrix. In mixed-species biofilms, the presence of a single species that secretes such enzymes would be beneficial for antibiotic-sensitive species in the same biofilm. Examples include *Moraxella catarrhalis* which secretes β-lactamases and, thereby, protects *S. pneumoniae* [81] and *H. influenzae* [82] from amoxicillin and ampicillin treatments, respectively, in mixed biofilms. 

To summarize, components of the ECM alter the biomass organization affecting diffusion of certain antibiotics within the biofilm and, thus, altering the level of exposition of cells located in specific biofilm niches particularly those within dense biomass. Additionally, certain ECM components can interact with antibiotics, preventing them from reaching their targets within cells. Furthermore, the ECM can retain antibiotic-modifying enzymes, which is particularly relevant in mixed biofilms, where susceptible bacteria can be protected by such enzymes released from resistant cohabitants. 

### 3.3. Physiological Heterogeneity

The architecture and organization of the biofilm also generates gradients of dispersion of nutrients, oxygen, pH, signaling molecules, and waste products. Oxygen and nutrient depletion occur in particular niches, such as inside the microcolonies and in the lower cell layers, and these conditions can induce a variety of physiological states involving different metabolism (aerobic, microaerobic, and fermentative) and growth rates (fast and slow growth, dormant cells, and persister cells) [83,84]. Nongrowing and slowly growing bacteria are less vulnerable to antibiotics as a consequence of the inactivity of antibiotic targets, a phenomenon called “drug indifference”. In contrast, persister cells are phenotypic variants that constitute a part of the population with tolerance to antibiotics (see Section 3.1) that can resume growth after antibiotic removal. Persisters are present in both biofilm and planktonic cultures; however, biofilms typically harbor more persisters than planktonic cultures [66]. Ultimately, the biofilm is constituted of cells with different physiological states and chances to survive an external drug insult. Indeed, this repertoire of physiological cell states is relevant for tolerance to multiple antibiotics. For example, slowly growing cells are tolerant to antibiotics such as tobramycin and ciprofloxacin [85], which target protein synthesis machinery and DNA gyrase, respectively, and thus, exert their activity on fast-growing cells. In contrast, slowly growing cells are susceptible to antibiotics such as colistin [86] that act on the membrane.

Bacteria respond to starvation and stress conditions through specific adaptative mechanisms, such as activation of the stringent response (SR) and the SOS response. The SR is induced by amino-acid, carbon, and iron starvation [87]. Under amino-acid deprivation, the ribosomes are stalled by the presence of uncharged tRNA in the A site (Figure 3A). (p)ppGpp synthetases, e.g., RelA and SpoT in β- and γ-proteobacteria, sense stalled ribosomes and synthetize (p)ppGpp, also known as *alarmone*, which initiates transcriptional reprogramming and regulates various metabolic pathways, such as phosphate, amino-acid and lipid metabolism, among others [87] (Figure 3A). (p)ppGpp also modulates the repressor CodY, a master regulator of many genes triggered during environmental stress. Ultimately, the SR shuts down almost all metabolic processes, and, thus, cells become tolerant to antibiotics that target such processes. For example, the SR in *E. coli* has been linked to tolerance to inhibitors of cell-wall biosynthesis, such as penicillins [88], cephalosporins and carbapenems [89], and of cell division, such as norfloxacin [90] and ofloxacin [91]. The SR has been related to reduced susceptibility to ofloxacin [92,93], meropenem, colistin [92], and gentamicin [92,94,95] in *P. aeruginosa* biofilms, and to tolerance to ampicillin and vancomycin in *S. aureus* [96]. 

The SOS response also contributes to antibiotic tolerance. It is generated by stress conditions such as DNA damage. Single-stranded DNA, generated by disruption of the DNA, activates RecA, which, in the presence of (d)ATP, stimulates self-cleavage of the repressor LexA leading to de-repression of SOS genes [97] (Figure 3B). Under regular physiological conditions, LexA is bound to a specific DNA sequence (SOS box) located upstream of several genes participating in DNA repair, mutagenesis and cell growth, and represses SOS gene expression (Figure 3B). The SOS response plays an important role in tolerance to antibiotics that cause DNA damage such as fluoroquinolones [98]. Studies in *E. coli* have demonstrated the function of the SOS response in enhancing biofilm tolerance to fluoroquinolones [91]. Together with the SR, the SOS response can also activate the expression of toxin-antitoxin (TA) modules [91,99], although TA modules can also be activated by stress-induced proteases like ClpXP and Lon in response to antibiotics. TA modules are genetic elements composed of two genes. One gene encodes a stable toxin protein that inhibits bacterial growth by interfering with essential cellular processes such as DNA replication, translation, cell wall synthesis, and membrane homeostasis, among others [100] (Figure 3C). The other gene encodes a cognate antitoxin that typically prevents or impairs toxin function. Antitoxins can be labile molecules that are degraded under stress conditions, a circumstance that allows the toxins to exert their harmful effects in cellular functions. Thus, by inactivating antibiotic targets, TA confers antibiotic tolerance [101] and increases persister formation [102] (Figure 3C). Several studies have shown upregulation of TA modules in persister cells. A well-studied example is MqsA and MqsR, a classical TA module where the MqsR functions as the toxin and MqsA as the antitoxin [103,104]. MqsR production is stimulated during biofilm formation and enhances cell motility in *E. coli* [103], and MqsA has been linked to the regulation of the general stress responses, such as oxidative stress [104]. MqsA represses the stress regulator RpoS, which decreases the concentration of the messenger 3,5-cyclic diguanylic acid and, consequently, biofilm formation is inhibited. However, upon stress conditions, such as oxidative stress, MqsA is unstable and rapidly degraded by Lon and ClpXP proteases, causing the accumulation of RpoS [104]. Then, SR is activated, and bacteria initiate biofilm formation. However, this has recently been disputed. Frainkin et al. (2019) reported that MqsA is not a global regulator and does not regulate *rpoS* expression. Moreover, authors showed that *mpsRA* production is not regulated by stress and that mutation of mpsRA has no clear effect on biofilm formation [105]. 

TAs are often involved in the stabilization of plasmids [106] and genomic islands that carry integrative and conjugative elements, which can mediate resistance to multiple antibiotics [107]. Considering that these genomic elements are commonly involved in promoting HGT [108], the role of TAs in antibiotic resistance can be notable. The SOS response and the SR participate in the dissemination of antibiotic resistance through integrons. Integrons are genetic elements involved in the capture antibiotic resistance genes. As they are located on mobile genetic elements, such as transposons, they contribute to the dissemination of these genes among Gram-negative bacteria [109]. An integron is composed of a gene encoding an integrase, a specific recombination site, and a promoter that controls the expression of promoter-less genes embedded within gene cassettes [110], some of which can be located in genetic mobile elements and contain antibiotic resistance-gene cassettes. Within the biofilm environment, transposases are activated under SOS response and SR regulates integrase expression [111], thereby promoting the dissemination of the antibiotic resistance genes located in mobile elements within members of the biofilm. 

Depletion of oxygen in the interior of biofilms [112,113] and the presence of oxygen gradients have been demonstrated in different biofilm models [114,115]. Hypoxia affects metabolism substantially, because it shifts aerobic respiration to alternative metabolic routes such as denitrification and fermentation. Consistent with the hypoxic conditions in biofilms is the detected production of nitrous oxide, an intermediate of denitrification, in sputum of cystic fibrosis patients with chronic *P. aeruginosa* infection [116]. Earlier work using 70 different Gram-negative and 23 Gram-positive bacteria growing in aerobic and anaerobic conditions showed that microorganisms were more tolerant to aminoglycosides and highly tolerant to tobramycin under anaerobic conditions [117]. Accordingly, *Pseudomonas* biofilms formed under anaerobic conditions were more tolerant to antibiotics such as tobramycin, ciprofloxacin, carbenicillin, ceftazidime, chloramphenicol, and tetracycline than those formed in aerobiosis [112]. Indeed, hypoxia reduces membrane potential conferring tolerance to antibiotics such as aminoglycosides that require an intact membrane potential to be transported into the cytoplasm [118].

Antibiotics can induce oxidative stress by increasing cellular hydroxyl radical levels and enhancing production of lethal reactive oxygen species (ROS) [119,120]. This could be caused by an increased oxidation rate of tricarboxylic-acid cycle-derived NADH that perturbs iron homeostasis. Ferrous iron from iron-sulfur clusters is oxidized to ferric iron in the Fenton reaction which yields extremely reactive and harmful hydroxyl radicals that oxidize vital macromolecules such as DNA, proteins, and lipids [120,121]. To counteract, bacteria stimulate production of catalases and superoxide dismutases. Indeed, *Pseudomonas* mutants lacking catalase generate biofilms more susceptible to ciprofloxacin than wild-type biofilms [122]. The SR increases catalase and superoxide dismutase levels and represses the production of 4-hydroxy-2-alkylquinolines, which are intercellular signaling molecules with prooxidant properties [92]. Additionally, persisters downregulate genes encoding proteins involved in the generation of ROS, including a ferredoxin reductase, which is involved in recycling Fe^3+^ to Fe^2+^ and thus drives the Fenton reaction, and upregulates genes involved in ROS detoxification. Thus, persister cells are, to some extent, protected against the detrimental effects of ROS produced upon antibiotic treatment. Considering that the SR and persister formation are more highly activated in biofilm cells than in planktonic cultures, biofilm cells can better deal with ROS induced by antibiotics.

### 3.4. Traffic of Substances across the Cell Envelope

Several proteins in the bacterial membranes function in the recognition and transport of substances, including antibiotics, into or out of the cell. This activity is facilitated by efflux pumps and porins (in Gram-negative bacteria) that mediate an active and passive transport, respectively. Efflux pumps can be divided into six families, e.g., the multidrug- and toxin-extrusion (MATE), small multidrug-resistance (SMDR), major facilitator (MF), ATP-binding cassette (ABC), resistance-nodulation-division (RND) and proteobacterial antimicrobial compound-efflux (PACE) families. These families display large differences concerning transporter structure, function, and substrate specificity and energy source [123]. All families use protein motif force for as driving force except for the ABC transporters, which use ATP hydrolysis, and some members of MATE family that use sodium gradient instead. 

The MATE family comprises proteins of 400–700 amino-acid residues organized in 12 α-helices [124]. These pumps participate in the extrusion of diverse antibiotics, such as ciprofloxacin, chloramphenicol, streptomycin, kanamycin, norfloxacin, and ampicillin [125]. Representative examples of this family in MDR bacteria are YdhE in *E. coli*, which transports kanamycin and acriflavin, amongst others [126], PmpM in *P. aeruginosa* [127], and AbeM in *A. baumannii* [128]. The SMR family is constituted of small proteins composed 100–120 amino-acid residues organized as homodimers with four transmembrane helices in each subunit. EmrE [129] of *E. coli* and Smr/QacC in *S. aureus* [130], both transporters of acriflavine, belong to this family. The MF family is constituted of membrane proteins with 400–600 amino-acid residues organized in 12–14 transmembrane α-helices. This family of transporters facilitates the passage of ions and carbohydrates across membranes, as well as antimicrobial agents such as tetracyclines and fluoroquinolones [131,132]. NorA, LmrS, and MdeA of *S. aureus* are well-known members of this family [133]. The ABC transporters are active transporters, often constituted of a transmembrane channel, formed by one or two proteins, and a dimeric cytoplasmic ATPase. The MacB transporter of *E. coli*, which operates in concert with the outer-membrane protein TolC and transports azithromycin, clarithromycin, and erythromycin, belongs to the ABC family [134]. Members of the PACE family transport acriflavine, proflavine, benzalkonium, acriflavine, and chlorhexidine [135]. Transporters of the RND family form a protein complex constituted of about 1000 amino-acid residues organized in a 12-helical structure in the membrane, but, in contrast to MF transporters, RND proteins possess large periplasmic domains. In Gram-negative bacteria, members of MF (e.g., EmrB), RND (e.g., MdtK), and ABC (e.g., MacB) families can be organized in a tripartite protein complex formed of an inner membrane protein, a membrane fusion protein, and an OM protein [123]. Collectively, this complex spans the entire cell envelope and allows for efficient excretion of antibiotics into the external medium. The AcrAB-TolC complex of *E. coli* [136], MexAB-OprM of *P. aeruginosa* [137], and AdeABC of *A. baumannii* [138], all RND family members that participate in tripartite transporters, have been implicated in bacterial biofilm resistance and biofilm formation.

Several lines of evidence relate efflux-pump production to biofilm formation and, directly or indirectly, to AMR/tolerance. First, some efflux-pump-encoding genes are upregulated in biofilms as compared to planktonic cells. This was detected in different transporter families. Examples are the *mdpF* gene of *E. coli*, which encodes a component of the MdtEF efflux pump (RND family) that participates in the tolerance to nitrosyl-mediated toxicity and that was upregulated in anaerobic conditions [139], a condition found in biofilms. Another example is the multidrug efflux genes *acrA* and *acrB* of *S.* Typhimurium [140]. Second, the exposition of cells to efflux-pump inhibitors reduces biofilm formation in several pathogens such as *E. coli* and *K. pneumoniae* [141], *P. aeruginosa* [142], and *S.* Typhimurium [143]. Third, mutants lacking known efflux systems exhibit a marked reduction in biofilm formation, for instance pump mutants in *Salmonella* showed reduced production of curli [143], which are implicated in biofilm formation. Fourth, several studies link efflux-pump production and acquired resistance of biofilms to antibiotics. Within *Pseudomonas* biofilms, MexAB-OprM and MexCD-OprJ are essential for resistance to azithromycin [144], and MexAB-OprM also mediates resistance to colistin [145]. Additionally, the ABC transporter encoded by the PA1874–1877 operon conferred protection to tobramycin in biofilms [146]. Biofilms of *E. coli* formed by mutants in genes that participate in the AcrAB-TolC system exhibited sensitivity to tobramycin, tetracycline, and the antiseptic benzalkonium, while mutants in the EmrAB system exhibited sensitivity to tobramycin [147]. Overall, these studies evidence that efflux pumps can directly contribute to expel antibiotics during biofilms that contribute to AMR but also ECM components that ultimately contribute to biofilm tolerance.

The OM of Gram-negative bacteria forms a barrier for both hydrophobic and hydrophilic solutes. Porins control the access of antibiotics from the environment to the periplasm. Porins are trimers of 16-stranded β-barrels located in the OM that provide selective access of small hydrophilic molecules to periplasm by diffusion through a water-filled channel present in each of the subunits [148]. Not surprisingly, MDR clinical isolates of *Enterobacteriaceae* often exhibit loss of porin production [149,150]. Several genetic mechanisms reduce or prevent porin synthesis, including downregulation of expression, premature stop codons or insertion elements. Besides, missense mutations can alter the permeability properties. Efflux pumps and porin production act in synergy and have been associated to biofilm production, particularly in *Enterobacteriaceae* [141,151]. In *K. pneumoniae*, the gene coding for the porin OmpK36 was downregulated and the *acrB* gene, coding for a component of a major multidrug-efflux pump, was upregulated in biofilms as compared to planktonic cells [152]. 

The production of efflux pumps and porins can be up- and downregulated, respectively, to reduce the intracellular accumulation of antibiotics, an adaptive phenomenon based on the transport regulation. Curiously, upregulation of the MDR transporter MdfA of *E. coli* not only makes cells resistant to the aminoglycosides neomycin and kanamycin but also increases their susceptibility to spectinomycin by an, as yet, unexplained mechanism [153,154]. In general, efflux pump production could be regulated by local and global transcriptional regulators, modulators, various substances, including antibiotics, small RNAs and two-component regulatory systems whose activation depends on environmental stimuli [155]. MarA, BrlR, SoxS, Rob, and AcrR are very well-known regulators of pump production in pathogenic bacteria [155]. Pump production can be regulated at multiple levels. AcrAB-TolC of *E. coli* is the best-studied example of regulation under a complex regulatory network. AcrAB synthesis is negatively regulated by the local repressor AcrR that represses *acrAB* expression [156]. In addition, the repressor AcrS regulates *acrAB* negatively [157], while the histone-like nucleoid structuring protein H-NS has a role in the network repressing the expression *acrS* [158]. Therefore, by negatively regulating *acrS*, H-NS is a positive regulator of *acrAB*. Furthermore, the two-component regulatory systems EvgAS and/or PhoQP activate *tolC* expression, while EvgAS also activates *acrAB* expression [159]. Finally, the global regulators SdiA [160], MarA, SoxS, and Rob activate expression of *acrAB*, and the latter three regulators also activate *tolC* and *micF* expression [155]. *micF* transcripts inhibit the translation of the mRNA of porin OmpF. As OmpF plays an important role in the influx of antibiotics, the bacterium thus controls the efflux and influx of antibiotics by activating AcrAB-TolC and abolishing OmpF production, respectively. Interestingly, while MarA upregulates pump production, it downregulates biofilm formation through activation of the *ycgZ-ymgABC* operon which eventually reduces curli formation [161]. Possibly, MarA helps to activate a mechanism for cells to escape from biofilms as defence to the antibiotic insult. However, since pump expression is regulated by multiple mechanisms, it is difficult to speculate about the precise biological role of MarA within the complex regulatory network. In *P. aeruginosa*, the production of the efflux pumps MexAB-OprM and EmrAB is differentially regulated by the regulators MdrR1 and MdrR2 [162]. These regulators activate EmrAB but repress MexAB-OprM. Their expression varies between cells located in different layers of a biofilm, which leads to different susceptibility to antibiotics dependent on the position within the biofilm biomass. As these efflux pumps have different substrate specificities, it has been proposed that MdrR1 and MdrR2 could act as master modulators controlling the activities of various pumps in different microenvironments within the stratified biofilm structure [162]. All together, these studies illustrate that the regulation of pump production can be controlled by a complex repertoire of regulators, some of which act on a variety of genes that participate in biofilm production, and/or by varying conditions, which are generated within the biofilm environment. 

### 3.5. Interbacterial Communication

QS is a population-density-dependent regulatory mechanism by which bacteria communicate via signaling molecules, called autoinducers. Bacteria produce autoinducers, which accumulate in the environment with the increase in the cell density. These autoinducers are recognized by cell-surface receptors or in the cytoplasm. After receptor recognition, gene transcription is activated, involving genes coding for surface proteins, transcription factors, virulence factors, and proteins involved in biofilm development [163,164]. Peptides are used as autoinducers in Gram-positive bacteria in contrast to the acylated homoserine lactones used in Gram-negative bacteria. Autoinducer 2 is used in Gram-negative and Gram-positive bacteria for intra- and interspecies communication.

QS seems to contribute to biofilm recalcitrance. Biofilms formed by QS mutants or wild-type bacteria treated with QS inhibitors are more susceptible to antibiotics. For example, *P. aeruginosa* biofilms formed by a mutant strain lacking *lasR* and *rhlR*, which is deficient in QS, were significantly more susceptible to tobramycin than wild-type biofilms [165]. Additionally, mixed *Pseudomonas* biofilms formed by QS mutants and wild-type bacteria exhibited a decreased resistance to tobramycin as compared to those formed only with wild-type bacteria [166]. In *S. aureus*, a QS-deficient *agrD* mutant exhibited a biofilm-specific decrease in resistance to rifampin compared to wild type [167]. Additionally, *fsrA* and *gelE* mutants of *E. faecalis*, which are deficient in QS and a QS-controlled protease, respectively, were impaired in biofilm formation in the presence of gentamicin, daptomycin, or linezolid, but not in the absence of these antibiotics [168]. The involvement of QS in biofilm recalcitrance may have multiple origins. QS is involved in biofilm formation and structuration; therefore, QS-defective mutants produce less structured biofilms. Considering that the architecture of the biofilm is relevant for its recalcitrant properties, the resulting biofilms would be more susceptible to antibiotics. Alternatively, QS may have other contributions. For instance, QS in *P. aeruginosa* regulates the production of 2-n-heptyl-4-hydroxyquinoline-N-oxide (HQNO), which inhibits the respiratory chain by binding to the cytochrome *bc*_1_ complex [169]. This results in the accumulation of ROS and the reduction in membrane potential and eventually in autolysis. Autolysis releases DNA, which, as previously discussed, promotes biofilm formation and confers resistance against positively charged antibiotics. Additionally, by reducing the electrochemical gradient, the sensitivity to aminoglycosides [170], tetracycline, and macrolides [171] is reduced. QS can also contribute to drug resistance within mixed species biofilms. *Stenotrophomonas maltophilia* and *P. aeruginosa* can form mixed biofilms when they coinfect cystic fibrosis patients. *P. aeruginosa* recognizes signal factors produced by *S. maltophilia* and induces the PmrAB two-component system that regulates resistance to cationic antimicrobial peptides [172]. Thus, QS signals and the resulting downstream consequences can elicit an ample range of physiological changes that alter the antimicrobial susceptibility of cells within a biofilm. 

Some intercellular communication requires direct cell-to-cell contact. Well-described examples are the two-partner secretion system (TPS). In the TPS system, a large surface-exposed protein, generically called TpsA, is secreted by an OM protein, called TpsB [173]. Some bacteria produce several TPS systems. TpsAs can have several functions in biofilm formation. They can function in adhesion to biotic surfaces and in interbacterial interactions [173,174]. In *N. meningitidis*, TpsA contributes to the maturation of biofilms, and its synthesis is upregulated during biofilm formation [175]. In many microorganisms, TpsA functions in inhibiting the growth of related bacteria [176,177]. In the proposed model, TpsA interacts with a conserved receptor on a target cell, after which a small C-terminal part is proteolytically released and transported into the target cell where it displays toxic activities [174]. Kin target cells produce an immunity protein that inhibits the toxic activity by specifically binding to the toxin. In this case, the imported toxin moiety functions as a signaling molecule and stimulates community associated behaviors, such as biofilm formation, as was demonstrated in *Burkholderia* [178]. Killed target bacteria release intracellular components, including DNA, which contributes to the biofilm formation. Additionally, this activity mediates resistance to cefotaxime in *E. coli* [179]. TpsA induces persister formation upon direct contact with cells lacking sufficient levels of immunity protein [179]. Very likely, more recently discovered secretion systems that deliver toxins to the target cells [180,181] also contribute to biofilm formation and recalcitrance, a research area that needs to be addressed. 

### 3.6. HGT in Biofilms 

HGT can involve the exchange of AMR genes between bacteria and is carried out through five different mechanisms. Three of them are generally known, namely conjugation (a direct transfer of genes between cells), transformation (acquisition of DNA from the environment), and transduction (gene transfer between cells via bacteriophages). The other mechanisms involve the release of membrane vesicles (MVs), which act as DNA reservoirs, or elongated membranous structures called nanotubes, which are employed for direct cell-to-cell contact. HGT can occur at a higher rate in biofilms than in planktonic cells [182]. Indeed, biofilms play an important role in the dissemination of AMR genes, and they are considered as reservoirs of resistance genes [10]. HGT would be favored within a biofilm for three main reasons: (i) the polymicrobial nature of biofilms that make them reservoirs of genetic diversity, (ii) the structure of biofilm, which restricts bacterial motility, increases cell density and promotes interbacterial interactions, and (iii) the presence of eDNA, which is released by cell lysis or by active secretion systems and that is retained in the ECM and establishes contacts among biofilm members. Probably even more important than its role as a glue for bacterial interactions, the eDNA can be taken up by transformation, one of the main HGT mechanisms. In addition, other factors involved in HGT would be the biomass surface, as it has been shown that high surface/volume ratios (in well-structured biofilms) increase the efficiency of plasmid transfer [183,184].

Conjugative plasmids and integrative and conjugative elements (ICEs) are transferred mostly via conjugation. Conjugation is carried out by a conjugation system based on sex pili that mediate direct contact between two cells, the donor and the recipient. After pilus retraction, intimate contact between donor and recipient allows for DNA transfer. This is probably the most common mechanism for the transfer of AMR genes within the biofilm environment. A study in *S. aureus* that showed a 16,000-fold higher transfer rate of the conjugative plasmid pGO1, which includes trimethoprim- and gentamicin-resistance genes, in biofilms than in planktonic cells, serves as a good example [185]. Likewise, in vitro biofilm experiments have demonstrated inter-family transfer by conjugation of a *bla*_NDM-1_ gene encoding a carbapenemase from *Enterobacteriaceae* into *P. aeruginosa* and *A. baumannii* [186]. This mechanism occurs more intensely in biofilms than in free-living bacteria because of the proximity between cells in this structure. Apart from conjugation, where cell–cell contact is established by pili, nanotubes can transport nonconjugative plasmids between closely related strains of *Bacillus subtilis* and of *E. coli* [187]. These structures have also been detected in MDR-related bacteria, such as *Acinetobacter baylyi* [188]. Future research will elucidate their involvement in HGT among biofilm members. 

Chromosomal DNA and nonconjugative plasmids are exchanged through transformation. Additionally, this mechanism is favored within biofilms because of the presence of large amounts of eDNA in the ECM. An experiment that compared the transformation rate in planktonic and biofilm cells of *N. gonorrhoeae* demonstrated that the transfer efficiency of two resistance genes, *ermC* and *aadA*, was higher at early stages of biofilm formation but decreased with biofilm age [189]. However, the transformation efficiency was shown not to depend on biofilm architecture. Spreading of transformants was observed in loose biofilms under selection pressure but was hardly observed from dense biofilms. Interestingly, even conjugative transposons of the Tn916 family, coding for tetracycline resistance, were shown to be transferred through this mechanism in in vitro grown biofilms of a multispecies consortium of oral bacteria [190].

Alternatively, genomic DNA, which may include AMR genes, can be transferred by transduction, although the diffusion of some phages through biofilms could be hampered by the biofilm matrix (discussed in Section 4). It can serve as an example, as a study showed that a temperate, Shiga-toxin (Stx)-encoding bacteriophage with a chloramphenicol-resistance gene (*cat*) inserted as a marker into the *stx* gene, could transfer this gene to *E. coli* within a biofilm [191]. Finally, although MV are released within biofilms and were demonstrated to transfer AMR genes, such as the β-lactamase-encoding *bla*_OXA-24_ gene in *A. baumannii* [192], their relevance in HGT in a biofilm environment remains to be studied. 

### 3.7. Mutation and Biofilms

Mutations in the bacterial genome can also give rise to AMR [193]. They may occur spontaneously, that is, in the absence of strong selective pressure. Spontaneous mutations are found to occur at a common rate of 10^−10^–10^−9^ per nucleotide per generation for many bacteria; therefore, mutants are usually already present as a minority within a population [194]. The mutation rate can increase significantly by exposure to agents that elicit oxidative stress. Oxidative stress is associated with the build-up of ROS, which may cause direct DNA damage and mutations. In certain circumstances, such as the exposure to sub lethal doses of bactericidal antibiotics, the accumulation of ROS is low, and it may promote resistance by the induction of the synthesis of multidrug efflux pumps and by mutagenesis [195]. This is a common situation when, for instance, subtherapeutic doses of antibiotics are used as growth promoters in animal production. However, more importantly, this may also occur within biofilms where antibiotic diffusion depends on the biofilm architecture. Additional mutations can also appear as a consequence of the SOS response, which induces the synthesis of error-prone DNA polymerases. Thus, stress responses can induce high mutation frequencies through different pathways. The mutation frequency may further increase after mutations are generated in the DNA failure-prevention or repair systems. The most frequent cause is related to defects in the methyl-directed mismatch repair system, e.g., in genes such as *mutS*, *mutL*, and *uvrD*. This can lead to a 100- to 1000-fold increase in the mutation rate [196,197]. The occurrence of microorganisms with this phenotype, called hypermutators, can represent an evolutionary advantage under selective pressure by increasing the possibility of acquiring favorable mutations, including mutations leading to AMR [198]. 

The hypermutator phenotype in biofilms has been detected in chronic infections in patients with cystic fibrosis, where 53% of the *Pseudomonas* isolates were hypermutable [199]. The frequency of mutants resistant to rifampicin and ciprofloxacin was higher in *P**seudomonas* biofilms than in free-living bacteria [200]. This state of hypermutability has also been reported in other bacteria isolated from cystic fibrosis patients, such as *S. aureus* and *H. influenzae* [201,202] but not in clinical isolates of the *Enterobacteriaceae* family from acute urinary tract infections governed by biofilms [203]. Thus, the hypermutation may be favored in some biofilm environments but is not a general mechanism. Altogether, bacteria in biofilms could be in a highly mutable state due to growth restrictions and nonlethal selective pressure, possibly further increased by antibiotic treatment, high- and hypermutability is also disadvantageous due to the accumulation of deleterious mutations. A solution could be transient hypermutability [204], where always a part of a population is transiently in a hypermutable state and prone to selection for favorable mutations, whereas the rest is rather stable. To the best of our knowledge, a transient hypermutator stage has not been experimentally demonstrated in biofilms. However, main conditions are accomplished: slow growth and nonlethal selective pressure. This raises the question whether transient hypermutability is a natural biological phenomenon that contributes to biofilm resistance. 

In general, mutations that affect the bacterial susceptibility to antibiotics were described (i) to alter an antibiotic target, (ii) to increase the production of efflux pump, (iii) to lead to changes in the cell membranes, or (iv) to increase the production or alter the substrate specificity of enzymes that inactivate antibiotics. For example, mutations affecting the antibiotic target of aminoglycosides were in the *rspL* gene [205], which code for 16S rRNA and the S12 ribosomal protein, respectively. Mutations in the *mexZ* gene in clinical isolates of *Pseudomonas* resulted in overproduction of the efflux system MexXY-OprM [206]. Colistin resistance has been associated with mutations in the genes coding for the PmrAB two-component regulatory system that regulates the addition of aminoarabinose to lipid A [207]. Additionally, mutations resulting in increased production of β-lactamases, e.g., by mutations in the promoter of the chromosomal *ampC* gene [208] or by increase of plasmid copy number [150] have been described, among others. In addition, the genes have evolved over the years, showing a large number of β-lactamase variants with point mutations in the gene resulting in changes in the amino-acid sequence [209]. This has led to the development of extended-spectrum β-lactamases (ESBLs) that degrade also first, second, and third generation cephalosporins and/or became resistant to β-lactamase inhibitors [210]. 

## 4. Control of Biofilm Infections

### 4.1. Lessons from Recalcitrant Mechanisms

As biofilms contribute to bacterial pathogenicity and recalcitrance, novel strategies and agents are required to deal with this issue. We have now clear evidence that the antibiotics used for the treatment of biofilm infections should be carefully selected, and such selection should consider the mechanisms of resistance and tolerance of biofilms. The use of cocktails of antibiotics would probably be more successful than a single antibiotic, but the antibiotic combination should also be thoroughly considered. Antibiotics should cover the heterogenic nature of biofilms. While one of the antibiotics in the combination should be active against persisters (e.g., colistin), others should target growing cells (e.g., ciprofloxacin, tobramycin, or β-lactams). In addition, the selection of antibiotics will benefit from the characterization of ECM composition, particularly the sorption and charge of the matrix, as these properties are relevant contributors to AMR. 

Many alternatives to antibiotics have been proposed to inhibit and/or eradicate biofilms. Their nature and their mechanisms of action are ample. In general, they possess one or several activities as (i) biofilm inhibitors, (ii) biofilm dispersers, and (iii) antimicrobials. An overview of these substances is listed in Table 1 and briefly discussed here. QS inhibitors can act as biofilm inhibitors or biofilm dispersers. Several plant-derived compounds exhibit this property, including halogenated furanones, which are molecules similar to N-acyl-homoserine lactones that prevent these QS signaling molecules to interact with their receptor, e.g., a LuxR family member. Thus, they function as antagonist of LuxR and repress expression of QS-induced genes [211,212]. Flavonoids, such as quercetin, can also interfere with QS. Quercetin represses the production of exopolysaccharides in *S. aureus*, required for initiation of biofilm formation [213,214]. However, QS involves a large variety of molecules in different organisms and their role in biofilm formation is species specific, thus, the activity of QS inhibitors is limited. Alternatively, enzymes that degrade QS signaling molecules such as lactonases that degrade lactone rings or phosphorylases have shown great promise [215,216], but again substrate specificity may limit their use. Other substances could contribute to inhibiting biofilm formation by interfering with the SR. For example, the 12-residue peptide 1018 interacts with (p)ppGpp and inhibits the accumulation of the alarmone and, thereby, persister formation, although this mechanism was later disputed [217]. The peptide prevents biofilm formation of different Gram-negative and Gram-positive bacteria [218] and revealed significant synergistic activity to eradicate biofilms in combination with antibiotics [219]. Eugenol is a secondary metabolite from clove (*Syzigium aromaticum*) with antibacterial activity. It inhibits biofilm formation and downregulates *relA* [220] leading to inhibition of the alarmone activation. 

ECM-disrupting enzymes are potentially also suitable inhibitors and dispersers of biofilms. Addition of exogenous enzymes such as Dispersin B or DNase I, which hydrolyze PNAG and eDNA, respectively, in combination with antibiotics eradicate biofilms of different species [290,291,292]. Other enzymes, such as alginate lyase [274], lysozyme [293], and lysostaphin [278], showed promising results in this respect. Additionally, proteases that cleave proteins of the ECM or proteins located at the bacterial cell surface with a function in biofilm formation disrupted streptococcal [279] and *S. aureus* biofilms [280]. Additionally, addition of exopolysaccharides of the ECM from biofilms of some bacteria can be used to inhibit biofilm formation of other microorganisms. For instance, Psl and Pel, which are produced in *Pseudomonas* biofilms, eradicate biofilms of *S. epidermidis* [289], and the polysaccharide A101 from *Vibrio* sp. QY101 disperses *Pseudomonas* biofilms [294]. Probably, these charged polysaccharides outcompete structures essential for biofilm integrity. Another molecule that destabilizes the ECM is ethyl pyruvate, which, in combination with the Ca^2+^-chelator EDTA, inhibits biofilms of many microorganisms [288].

Molecules that inhibit the adhesion properties of bacteria prevent the initiation of biofilm formation. Mannosides are small molecules that inhibit FimH [287], a mannose-binding component of the Type I pili that facilitates adhesion of uropathogenic *E. coli*. Mannosides can be used in combination with antibacterial agents to prevent biofilms on catheters [295]. Similarly, pilicides, which are small ring-fused 2-pyridones, inhibit Type I piliation [286]. Small peptides, such as FN075 and BibC6, block the assembly of curli and pili by disrupting protein–protein interactions during assembly and thereby inhibit the formation of *E. coli* biofilms [285]. Overall, biofilm inhibitors and dispersers utilize different mechanisms that ultimately disrupt intermolecular interactions required for the biogenesis and establishment of biofilms or they degrade these components. As these activities do not affect bacterial viability, they must be provided in combination with antimicrobials for bacterial eradication. 

### 4.2. Antimicrobial Substances

New antimicrobial substances, some of which exhibit good penetration in biofilms, have been proposed as alternatives to antibiotics. Among them, antimicrobial peptides, bacteriophages, and essential oils stand out as most promising and several examples are listed in Table 1. Antimicrobial peptides (AMPs) are small peptides of about 12–50 amino-acid residues, containing a considerable number of hydrophobic residues (≈50%) and positively charged residues [296]. They are produced by the innate immune system of animals, insects, plants, and humans to prevent bacterial, fungal, and viral infections [297,298]. They disrupt bacterial membranes through either one of three different mechanisms, (i) detergent-like membrane packing disruption, (ii) formation of pores in the barrel-stave model and (iii) toroidalpore model [299]. In addition, they can inhibit DNA, RNA, and protein synthesis. Hence, AMPs have a broad activity spectrum against microbes and, consequently, the probability of AMR development is relatively low compared to conventional antibiotics. Some AMPs from different sources have shown a good combination of antimicrobial and antibiofilm activities against superbugs (see examples in Table 1 and expanded in the AMP database: http://aps.unmc.edu/AP). An example is melittin, which is a major component of honeybee venom [300]. This cationic linear peptide of 26 amino-acid residues inserts into bacterial membranes forming short-lived pores and it also inhibits biofilm formation of several bacteria, including *P. aeruginosa* [221,223] and *K. pneumoniae* [222], among others (Table 1). However, natural AMPs exhibit drawbacks for their application in vivo, including low efficiency, low biostability due to enzymatic degradation, toxicity at the required concentrations, and inefficient delivery to the infection niche. In an effort to improve their utility, several strategies are being conducted, comprising the design of synthetic AMPs, combination with antibiotics, or conjugation to carriers (Table 1). As an example, cyclic derivatives of peptide1018 have been created to enhance the proteolytic stability and reduce aggregation of the peptide [301]. When 1018 was coadministrated with antibiotics, a high synergistic ability to prevent and eradicate biofilms of many bacteria was observed [219]. Some AMPs were encapsulated in vehicles such as polymers, nanoparticles, micelles, carbon nanotubes, and others [reviewed in 302]. The AMPs-carrying vehicles can diffuse through tissue layers and expose simultaneously a large number of peptides improving their effectivity while lowering toxicity and reducing degradation. One of the most commonly used delivery systems is gold nanoparticles (AuNPs) [302]. They have been proposed as conjugate to AMPs because they are of small size, high solubility, stability, and biocompatibility. Esculentin-1a conjugated to AuNPs [AuNPs@Esc(1–21)] exhibited about 15-fold higher activity than the peptide alone and, in contrast to the peptide, it was not toxic. In addition, it showed high resistance to proteases [238]. 

Phage therapy involves the use of lytic bacteriophages to kill bacteria [303]. It has some advantages compared with other antimicrobials, for example, their natural origin, lack of toxicity for humans or nontarget microbes, and their effectiveness against antibiotic-resistant bacteria. Moreover, they are self-replicating in the presence of host cells and disappear without host. As a disadvantage, phages are strain specific; hence, a successful treatment requires a full understanding of bacteriophage–host interactions, involving identification of the specific phage. Although the chance seems to be low, bacteria can acquire phage resistance at high frequency. A simple point mutation in the phage receptor on the bacterial cell surface already suffices for the bacteria to escape phage attack. In addition, bacteria have several broad strategies to escape from phage, e.g., CRISPR-Cas, restriction enzymes, O-antigen, etc. To overcome these limitations, a combination of phages (phage cocktails) is often recommended instead of a single phage. This therapy is already supported by authorities in certain countries where commercial products against bacteria, such as *Listeria*
*monocytogenes*, *Salmonella enterica* and *E. coli*, as surface disinfectants or processing aids are available. Yet, while phage receptors are fully available in planktonic cells, their accessibility in biofilms is compromised. ECM structures can establish electrostatic interactions with phage particles preventing them from reaching the cell surface. However, some phages may carry polysaccharide-degrading enzymes and thus gain access to receptors on the bacterial cell wall. Additionally, the ECM contains released phage receptors from cell lysis that ultimately compete with cell-surface receptors. Enzymes contained in ECM, such as proteases, inactivate phages. On the other hand, the architecture of biofilms may limit phage diffusion. Biofilms with dense cell clusters established by tight cell–cell binding can limit the access of the phage to the entire community. Dormant cells within the biofilm are less susceptible to phages, as phage replication requires active bacterial metabolism [304,305]. Furthermore, biofilms can generate a state of hypermutability that stimulates the occurrence of phage resistance. Indeed, the emergence of phage-resistant populations among bacteria after phage therapy has been reported [251,306]. Overall, although lytic phages have bactericidal activities, only some hold some promise in the treatment biofilm infections (Table 1). For example, phage EFDG1 showed success in eliminating biofilms in vitro and preventing infection by *E. faecalis* and *E. faecium* [240], and bacteriophage vB_EfaH_EF1TV was recently shown to kill clinical *E. faecalis* strains and disrupt their biofilms [241]. Yet, to overcome phage-therapy limitations, different strategies are currently followed including combination with antimicrobials, phage cocktails, and genetically manipulated phages (Table 1). Examples of the latter include phages producing biofilm degrading enzymes such as dispersin B [307] or inhibiting enzymes or enzymes that that contribute to antibiotic penetration such as OmpF porin to enhance antibiotic penetration [308]. 

Essential oils extracted from plants comprise complex mixtures of volatile substances, including terpenes, terpenoids, and phenols, among others. Some of these compounds possess antimicrobial activity as they constitute part of the immune defense mechanism of plants against infectious agents. Several studies reported the antibiofilm activity of some essential oils against bacteria (Table 1). Many of them damage the bacterial membranes leading to the release of cytoplasm, although their mechanism of action is not uniquely caused by this route. Other essential oils also regulate the expression of genes involved in biofilm formation and biofilm dispersal. For example, essential oils from thyme, cinnamon, and clove exhibited a high antibiofilm activity against many bacteria, including ESKAPE bacteria [257]. Cinnamon oil was earlier proven to inhibit the production of rhamnolipids, proteases, and alginate as well as swarming motility in *P. aeruginosa* [256], which is consistent with inhibition of QS. Another essential oil, tea tree oil (TTO), has shown antibacterial and antibiofilm activity. TTO eradicates *S. aureus* biofilms by affecting the expression of 304 genes participating in many metabolic routes [260] and regulators such as SarA. The global regulator SarA positively controls expression of genes involved in biofilm formation. Additionally, the expression of *cidA* that encodes a murein-hydrolase regulator was downregulated whereas the expression of the *lgrA* and *B* operons, which inhibit autolysis, was upregulated. Together, this reduces the release of eDNA, which is a key component of the *S. aureus* ECM. Other studies have investigated the synergistic action of essential oils or their active principles with other antimicrobial molecules such as synthetic antimicrobial polymers (Table 1). Thus, these and other studies demonstrate that essential oils have a repertoire of killing activities and that they are promising as treatments against biofilms. Yet, their extraction is one of the most effort-requiring and time-consuming processes, which increases the costs of their application.

### 4.3. Alternative Methods 

Physical methods hold promise for eradication and inhibition of biofilms. This is particularly important on surfaces such as chronic wounds [309,310], infected prosthetics, implants, and medical devices. Good examples of these methods include nanoparticles, sonication, irradiation (ultraviolet, visible, or infrared light), or biomaterials. Indeed, blue light, for example, was effective against biofilms formed by *A. baumannii*, *P. aeruginosa*, and *N. gonorrhoeae* although less so against biofilms of *E. coli* and *E. faecalis* [311]. In vivo studies in mouse burns showed that blue-light exposure could drastically reduce bacterial load and effectively protect mice from lethal infection with *P. aeruginosa* [312]. Blue light presumably exerts its effect on bacterial cells by exciting porphyrins which then generate ROS, as suggested by the resistance to blue light exhibited by a *P. aeruginosa* mutant defective in porphyrin biosynthetis [313]. Additionally, ultraviolet C light has been shown to efficiently eradicate *Pseudomonas* biofilms on precontaminated catheter-like tubes [314]. In general, irradiation has only an effect on superficial epidermal layers or the surface of materials because of poor accessibility of deeper tissues. On the other hand, nanoparticles, prepared from diverse materials, including both organic and inorganic materials, have a broad spectrum of antibacterial and antibiofilm activities, e.g., by disrupting bacterial membranes, interacting with proteins or DNA, or promoting the production of ROS [315]. Alternatively, different materials with topographic patterning have been developed to prevent biofilm formation. As the nature, hydrophobicity, and topology of the materials are relevant for substrate–bacteria interactions, these characteristics are conveniently modified in biomedical polymeric surfaces to generate catheters, implants, or devices with reduced bacteria-binding capacity [316,317]. Their effectivity is enhanced in combination with other antibacterial strategies, e.g., the use of nitric oxide-releasing materials, which, together, showed high synergic activity in the inhibition of bacterial growth and biofilm formation of *S. epidermidis* [318]. To summarize, compared with conventional antimicrobial agents, physical methods exhibit a broad-spectrum effectiveness under ambient conditions, are easy to operate at low cost, and low maintenance. Important disadvantages are long exposition times, and their application is mostly restricted to surfaces. Another interesting strategy for controlling biofilms is the use of probiotics, e.g., live microorganisms with demonstrated health benefits that inhibit pathogenic biofilms. Probiotics have been considered for human therapeutic applications, and even bacterial strains have been genetically modified to kill pathogenic strains and inhibit biofilm production. Probably the best example is the *E. coli* strain Nissle 1917 that has been extensively used for treatment of intestinal disorders [319]. This strain inhibits biofilm formation of pathogenic and nonpathogenic *E. coli*, *S. aureus*, and *S. epidermidis* [320]. In an attempt to improve its therapeutic potential, the strain was genetically modified to synthesize an antibiofilm enzyme, dispersin B, in response to the detection of autoinducers secreted by *P. aeruginosa*. The recombinant strain was active against *P. aeruginosa* gut infection in animal models [321]. 

## 5. Concluding Remarks

The capacity of microorganisms to evolve and adapt to environmental cues has led to a health crisis as they became resistant to most, or almost all, commercial antibiotics. Biofilm formation is an ancient form of bacterial adaptation that contributes substantially to the problem because of their recalcitrance to treatment. Indeed, biofilms are the origin of significant morbidity and mortality. As discussed here, biofilm recalcitrance integrates many mechanisms, including metabolic heterogeneity, stress responses, efflux pump regulation, entrapment and inactivation of antibiotics in the ECM, interbacterial communication, increased mutability, and exchange of genetic material. Many of these factors have been discovered particularly in strains of *P. aeruginosa*. However, the specificity and multifaceted nature of the described mechanisms indicate the necessity of studying them also in other bacteria. Even more challenging, but necessary, will be to study biofilms in natural infections, where heterogeneous bacterial populations are common, and many environmental factors, including host defenses or diffusion of antibiotics in tissues, are present.

The understanding of the mechanisms that mediate recalcitrance will definitely guide therapeutic strategies to successfully deal with biofilm infections. These should be accompanied with methodologies for rapid diagnosis of biofilm infections and characterization of the biofilm biology and composition in vivo. Additionally, the availability of a panel of substances to inhibit and disperse biofilms will contribute to the selection of adequate therapeutic strategies to deal with particular biofilm infections. 

## Figures and Tables

**Figure 1 antibiotics-10-00003-f001:**
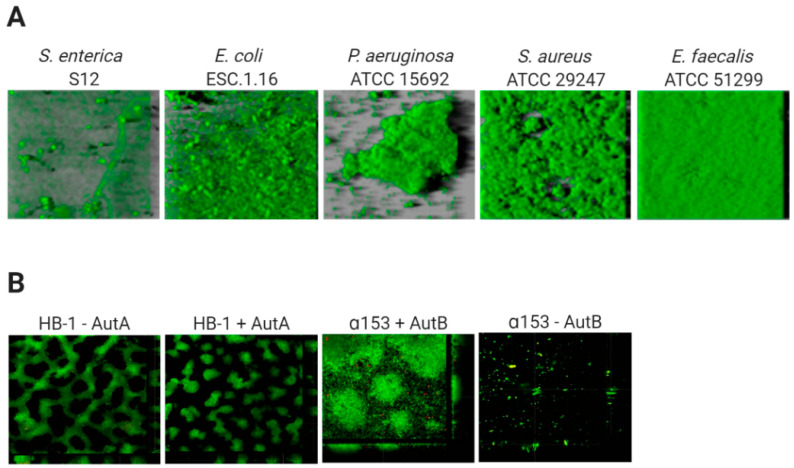
Variable architecture of biofilms. (**A**) Biofilms of five species (*Salmonella enterica*, *Escherichia coli*, *Pseudomonas aeruginosa Staphylococcus aureus*, *Enterococcus faecalis)* were formed under static conditions on abiotic surfaces during 24 h and were stained with Syto9, a green fluorescent nucleic acid marker. Reprinted from [51] with permission from Elsevier. (**B**) Strains of *Neisseria meningitidis* HB-1 and α153 and derivatives, which do or do not produce the autotransporters AutA and AutB (as indicated), formed biofilms under flow conditions during 14 h and were stained with the LIVE/DEAD Backlight bacterial viability stain (where red cells are dead and green cells are live). Reproduced from [52,53].

**Figure 2 antibiotics-10-00003-f002:**
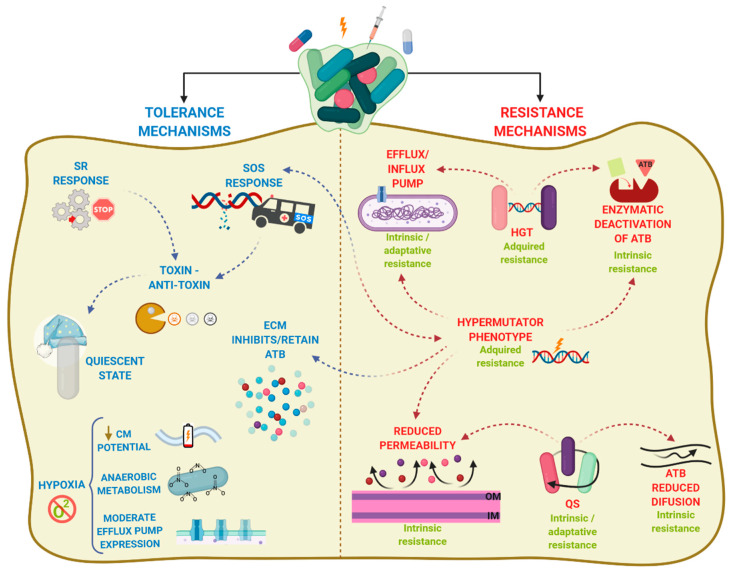
Mechanisms of biofilms recalcitrance. Biofilm recalcitrance comprises a combination of antibiotic (ATB) resistance mechanisms (right panel) and ATB tolerance mechanisms (left panel). Resistance mechanisms confer the ability to survive and grow at increased ATB concentrations for long periods and involve horizontal genetic transfer (HGT), hypermutation, and quorum sensing (QS), leading to transport of antibiotics via efflux pumps, reduced permeability of the outer membrane, or production of enzymes that inactivate ATB. The type of AMR (Box 1) is indicated in green. In contrast, tolerance mechanisms lead microorganisms to survive at increased ATB concentrations temporally, and involve activation of stress responses (SOS response, stringent response SR)) and hypoxia, leading to activation of a quiescent state, anaerobic metabolism, decrease of membrane potential, and moderate increase in efflux pump expression. Arrowhead lines indicate the interrelation between mechanisms. CM: cytoplasmic membrane.

**Figure 3 antibiotics-10-00003-f003:**
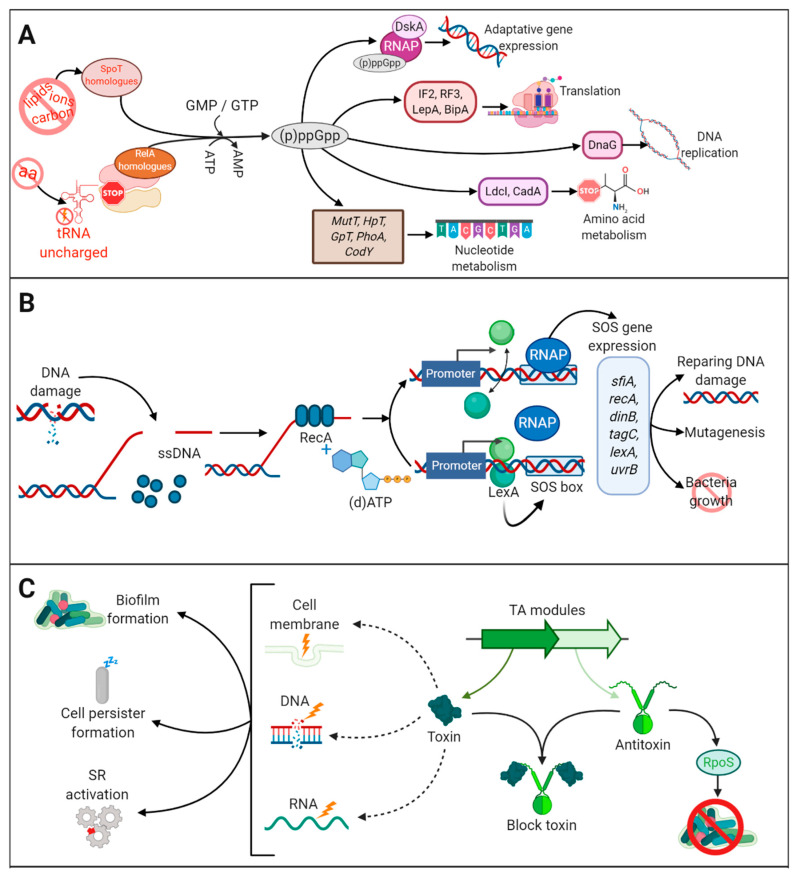
The stringent response, toxin-antitoxin, and SOS response pathways. (**A**) The stringent response (SR) is triggered by several stress conditions (amino-acid, carbon and iron deprivation or membrane damage) that activate the production (p)ppGpp by the synthetases RelA and SpoT and homologues. (p)ppGpp reprograms cell metabolism through the interaction with proteins involved in translation, transcription, replication, amino-acid metabolism, and nucleotide metabolism. (**B**) The SOS response is triggered by damaged DNA. Single-stranded (ssDNA) is detected by RecA. In the presence of (d)ATP, RecA is activated causing self-cleavage of LexA. LexA is a dimer that represses the transcription of SOS genes, which harbor an SOS box in their promoter. Cleavage of LexA leads to activation of the SOS genes inducing a repertoire of activities as indicated. The *lexA* gene also contains an SOS box; therefore, LexA production is self-regulated. Once the DNA is repaired, LexA represses the SOS response. (**C**) Several toxin-antitoxin (TA) modules are dispersed on the chromosome and are further classified according to the nature of the antitoxin and its mechanism of action. They are constituted by two genes, one encoding a toxin, with specific activities against target molecules (DNA, RNA, membrane, cell wall synthesis, ATP), and the other an antitoxin that binds to the toxin and inhibits its toxic activity. Under normal conditions, toxin and antitoxin are equally produced and thus the toxin does not exert its function. However, under stress conditions, the antitoxin can be degraded, and the toxin is free for toxic reactions. SR and SOS responses increase the production of ClpXP and Lon proteases, which activates the toxin by degradation of the antitoxin TA modules production. The toxic activity has several repercussions on cell biology.

**Table 1 antibiotics-10-00003-t001:** Proposed alternatives to antibiotics with antimicrobial or antibiofilm activities. The substance, the mechanism of action (including anti-biofilm activity), and the target bacterial species are indicated for each agent.

Substance(s)	Mechanism of Action	Targets	References
***Antimicrobial Peptides***			
*Natural Antimicrobial Peptides*			
Melittin	Formation of short-lived pores in the membrane and increase of permeability of OM	*P. aeruginosa*, *S. aureus*, *E. coli*, *K. pneumoniae*, *A. baumannii*	[220,221,222,223,224]
Japonicin-2LF	Detergent-like activity against components of biofilm matrix; higher activity in inhibiting than in eradicating biofilms	*S. aureus*, MRSA, *E. coli*	[225]
Magainin 2	Destabilizes the bacterial membrane and intracellular processes	*A. baumannii*, *P. aeruginosa*, *E. coli*	[226,227,228]
LL-37	Membrane disruption; inhibits twitching and QS; interferes in bacterial attachment; downregulates *rhlA* and *rhlB* genes	*P. aeruginosa*, *A. baumanni*, *S. aureus*	[229,230,231]
Temporin 1Tb	Disruption of cell membrane integrity; capable of penetrating biofilm and killing bacteria; hemolytic activity	*S. epidermidis*, *S. aureus*, *K. pneumoniae*, *P. aeruginosa*, *E. faecium*	[232,233]
*Synthetic Antimicrobial Peptides*			
1037	Downregulates genes of biofilm development; reduces swimming and swarming motilities	*P. aeruginosa*, *L. monocytogenes*, *Burkolderia cenocepacia*	[218]
Esculentin (1–21)	Biofilm eradication	*P. aeruginosa*	[234]
1018	Binds (p)ppGpp and inhibits SR; inhibits attachment, QS, and twitching motility	*E. coli*, *S. aureus*, MRSA, *P. aeruginosa*, *A. baumannii*, *K. pneumoniae*, *A. baumannii*, *S.* Typhimurium, *E. faecium*	[218,219,235]
STAMP G10KHc	Disrupts and permeabilizes OM and IM	*P. aeruginosa*	[236]
F_2,5,12_W	Reduces initial adhesion of bacteria; eliminates mature biofilms; suppresses biofilm formation	*S. epidermidis*	[237]
*Combined Therapies*			
1018 + antibiotics (e.g., ciprofloxacin)	Inhibition of (p)ppGpp activation; downregulation of genes that interfere with antibiotic resistance and biofilm formation	*E. coli*, MRSA, *P. aeruginosa*, *K. pneumoniae*, *A. baumannii*, *S. enterica*	[219]
Esculentin (1–21) + AuNPs (AuNPs@Esc(1–21))	Disruption of membrane forming clusters	*P. aeruginosa*	[238]
Temporin 1Tb + EDTA	Mature biofilm eradication	*S. epidermidis*	[232]
lin-SB056-1 + EDTA	Perturbation of membrane; eradication biofilm; chelation of divalent metal ions	*P. aeruginosa*	[239]
***Bacteriophages***			
*Phages*			
EFDG1	Mature biofilm eradication	*E. faecium*, *E. faecalis*	[240]
vB_EfaH_EF1TV	Mature biofilm eradication	*E. faecalis*	[241]
vB_PaeM_LS1	Disrupts and avoids dispersion of biofilms; inhibits biofilm growth	*P. aeruginosa*	[242]
vB_SauM_philPLA-RODI	Penetrates biofilms; inhibits biofilm formation	*S. aureus* *S. epidermidis*	[243]
*Phage-derived Enzymes*			
LysAB3	Degradation of bacterial wall peptidoglycan, biofilm eradication	*A. baumannii*	[244]
Dpo48	Degrades exopolysaccharide and eradicates biofilm	*A. baumannii*	[245]
*Combined Phage Therapy*			
Phage + amoxicillin	Biofilm eradication	*K. pneumoniae*	[246]
SAP-26 + rifampicin	Hydrolysis of bacterial wall; mature biofilm eradication; reduction of biofilm growth	*S. aureus*	[247]
Phage K + DRA88	Inhibits biofilm formation; disperses biofilms	*S. aureus*	[248]
Phage K + its derivatives (e.g., K.MS811)	Biofilm eradication	*S. aureus*	[249,250]
Phage M4 + E2005-24-39 + E2005-40-16 + W2005-24-39 + W2005-37-18-03	Biofilm eradication	*P. aeruginosa*	[251]
DL52 + DL54 + DL60 + DL62 + DL64 + DL68	Attachment to cell by binding to lipopolysaccharide; biofilm eradication	*P. aeruginosa*	[252]
***Plant-Derived Natural Products***			
*Essential Oils or Principal Active Compounds*			
Cinnamon (cinnamaldehyde)	Inhibits QS mechanism: regulates production of rhamnolipids, proteases, and alginate and swarming activity; disrupts synthesis of DNA, RNA, proteins, lipids, and polysaccharides; alters expression of genes related to biofilm formation (e.g., *icaA*)	*E. coli*, *P. aeruginosa*, *K. pneumoniae*, *A. baumannii*, *S. epidermidis*, *S. aureus*, MRSA, *S. enteridis*, *S.* Typhimurium	[253,254,255,256,257]
Clove	Disrupts QS communication: biofilm dispersal, inhibits AHL synthesis; downregulates *relA* gene	*E. coli*, *P. aeruginosa*, *K. pneumoniae*, *A. baumannii*, *S. aureus*	[253,257]
Thyme (thymol)	Downregulates *sarA* gene; increases membrane permeability; penetrates polysaccharide matrix: eradicates biofilms	*E. coli*, *P. aeruginosa*, *K. pneumoniae*, *A. baumannii*, *S. aureus*, *S. enteridis*	[257,258,259]
Tea tree oil	Alters expression of multiple genes related to biofilm formation (e.g., *sarA*, *cidA*, *igrA*, *ifrB*)	*S. aureus*	[260]
Oregano (carvacrol)	Increases membrane permeability; penetrates polysaccharide matrix; eradicates biofilms	*K. pneumoniae*, *P. aeruginosa*, *A. baumannii*	[258]
Halogenated furanones	QS inhibition; antagonist of LuxR	*E. coli* *P. aeruginosa*	[211,212]
Flavonoids (e.g., quercetin)	Represses exopolysaccharides production; inhibits *rpoS* gene expression; decreases swimming motility	*S. aureus*, *E. coli*, *P. aeruginosa*, *E. faecalis*	[261,262,263,264]
*Combined Therapy*			
Carvacrol + eugenol	Increases membrane permeability	*K. pneumoniae*, *P. aeruginosa*, *A. baumannii*, *S. aureus*	[258,265,266]
Cinnamaldehyde + eugenol	Membrane permeabilization	*S. epidermidis*	[267]
Curcumin + antibiotics (e.g., ciprofloxacin)	QS inhibition	*E. coli*, *K. pneumoniae*, *P. aeruginosa*, *S. aureus*, *E. faecalis*	[268]
*Enzymes*			
Dispersin B	Hydrolyses PNAG	*S. epidermidis*, *S. aureus*, *E. coli*, *A. pleuropneumoniae*	[38,269,270]
DNases	Hydrolyses DNA	*A. baumannii*, *K. pneumoniae*, *E. coli*, *P. aeruginosa*, *S. aureus*	[271,272,273]
Alginate lyase	Degrades alginate	*P. aeruginosa*	[274]
Lysozyme	Hydrolytic activity	*S. pneumoniae*, *Gardnerella vaginalis*, *S. aureus*, *P. aeruginosa*	[275,276,277]
Lysostaphin	Degrades cell wall	*S. aureus*, *S. epidermidis*	[278]
Proteases (e.g., SpeB)	Degrades cell wall	*Streptococcus spp.* *P. aeruginosa*, *S. aureus*	[279,280]
Paraoxonases (e.g., acylase I)	Inhibits QS	*A. hydrophila*, *P. putida*, *P. aeruginosa*	[216,281,282]
Lactonase	Inhibits QS	*P. aeruginosa*	[215,283]
*Small molecules*			
Small molecules (e.g., LP 3134, LP 3145, LP 4010)	Inhibition of diguanylate cyclase	*P. aeruginosa*, *A. baumannii*	[284]
Pilicides (FN075, BibC6, Ec240)	Blocks synthesis of curli and Type I pili, and inhibits chaperone-usher pathway for pili biogenesis	*E. coli*	[285,286]
Mannosides	Inhibits FimH of type I pili	*E. coli*	[287]
Ethyl pyruvate	Inhibits enzymes of the glycolytic pathway	*E. coli*	[288]
*Polysaccharides*			
Psl, Pel	Disperses biofilm	*S. epidermidis*	[289]

OM: outer membrane; MRSA: methicillin-resistant *S. aureus*; QS: quorum sensing; SR: stringent response; STAMP: selectively targeted antimicrobial peptide; IM: inner membrane; EDTA: ethylenediaminetetraacetic acid; AHL: *N*-acyl-homoserine actones; PNAG: poly-(β-1,6)-*N*-acetylglucosamine; DNase: deoxyribonuclease; ECM: extracellular matrix.

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
