# Peer review of "Biofilms as Promoters of Bacterial Antibiotic Resistance and Tolerance"

_antibiotics, 2020, doi:10.3390/antibiotics10010003_

Round 1

Reviewer 1 Report

The manuscript by Gema et al “Biofilms as promoters of antibiotic resistance an tolerance in mult-drug resistant bacteria” summarizes the antibiotic-resistant or tolerant mechanisms of multi-drug resistant biofilm cells. This manuscript provides very detailed and diverse recalcitrant mechanisms of biofilms to antibiotics from ECM, Physiological heterogeneity, cell envelope components, interbacterial interactions, HGT, and mutations. According to the recalcitrance mechanisms, strategies to control biofilms were also summarized. This review provides insights to understand MDR biofilms and develop effective strategies to control MDR biofilms.  

I have a couple of suggestions to improve this comprehensive manuscript.

Recently, there has been a discussion about distinguishing tolerance and persistence. Tolerant cells are eventually killed by antibiotics but slowly compared to the susceptible cells, while persister cells survived a prolonged time by showing a biphasic survival curve. See the following reference.

Balaban, N.Q., Helaine, S., Lewis, K., Ackermann, M., Aldridge, B., Andersson, D.I., et al. (2019) Definitions and guidelines for research on antibiotic persistence. Nat. Rev. Microbiol. 17: 441–448.

Recently, protein engineering and synthetic biology have been applied to control biofilm formation. It would be good to include such efforts in this review.

Minor errors/suggestions:

 In the Abstract, MDR was used without abbreviation from the full name.

Be consistent using one between “Multi-Drug Resistant bacteria “and “multidrug resistant bacteria.”

L158, typo error, Exopolysacharides --> Exopolysaccharides,  and PsI (upper case i was used) --> Psl (lowercase L)

Remove “,” in ATCC numbers in L206 and 207

There is no section 2 title. Between 1 introduction and 3  Mechanisms of biofilm recalcitrance.

L286, pH non-italic

L482, isra gene --> lsrA gene?

L709 and L710, gen --> gene

L711, Colistine --> Colistin

L812, micells --> micelles

L831, Lysteria --> Listeria

Author Response

The manuscript by Gema et al “Biofilms as promoters of antibiotic resistance an tolerance in mult-drug resistant bacteria” summarizes the antibiotic-resistant or tolerant mechanisms of multi-drug resistant biofilm cells. This manuscript provides very detailed and diverse recalcitrant mechanisms of biofilms to antibiotics from ECM, Physiological heterogeneity, cell envelope components, interbacterial interactions, HGT, and mutations. According to the recalcitrance mechanisms, strategies to control biofilms were also summarized. This review provides insights to understand MDR biofilms and develop effective strategies to control MDR biofilms.  

I have a couple of suggestions to improve this comprehensive manuscript.

 Recently, there has been a discussion about distinguishing tolerance and persistence. Tolerant cells are eventually killed by antibiotics but slowly compared to the susceptible cells, while persister cells survived a prolonged time by showing a biphasic survival curve. See the following reference.

Balaban, N.Q., Helaine, S., Lewis, K., Ackermann, M., Aldridge, B., Andersson, D.I., et al. (2019) Definitions and guidelines for research on antibiotic persistence. Nat. Rev. Microbiol. 17: 441–448.

Re: We agree. We have expanded both definitions in the section 3.1.

Recently, protein engineering and synthetic biology have been applied to control biofilm formation. It would be good to include such efforts in this review.

Re: Works in protein engineering and synthetic biology are discussed through section 4. We have not generated a novel section dedicated to this topic because i) there is a recent review (Fang et al, 2020), and ii) the section 4 is organized in antimicrobial substances (4.1), anti-biofilm substances (4.2) and other alternatives (4.3) where works in protein engineering and synthetic biology belongs to the three sections.

Minor errors/suggestions:

 In the Abstract, MDR was used without abbreviation from the full name.

Be consistent using one between “Multi-Drug Resistant bacteria “and “multidrug resistant bacteria.”

L158, typo error, Exopolysacharides --> Exopolysaccharides,  and PsI (upper case i was used) --> Psl (lowercase L)

Remove “,” in ATCC numbers in L206 and 207

There is no section 2 title. Between 1 introduction and 3  Mechanisms of biofilm recalcitrance.

L286, pH non-italic

L482, isra gene --> lsrA gene?

L709 and L710, gen --> gene

L711, Colistine --> Colistin

L812, micells --> micelles

L831, Lysteria --> Listeria

Re: thanks to point it out these mistakes. We have reviewed the entire manuscript and considered all the suggestions.

Reviewer 2 Report

-There are several sentences throughout the manuscript that are identical to the existing literature and raises concern about overall syntax and writing of the review manuscript. There are also more than one font used throughout the manuscript. 

-Abstract of the study is not providing the results of comprehensive analysis and review of the literature and simply state the problem rather than discussing outcomes derived from this review study. 

-There are major and important microbiological errors in the manuscript, for example, authors indicate Salmonella spp on line 29 that is not a correct nomenclature. This genus is now has only 2 species (bongri and enterica) and S. bongori is not a human pathogen so all serovars of Salmonella are derived from one species, hence Salmonella spp. is incorrect terminology and should be replaced by Salmonella serovars. This is a common mistake in the literature and CDC does a great job explaining this at: 

https://wwwnc.cdc.gov/eid/page/scientific-nomenclature  

-Key references are not discussed, although authors discuss a WHO report, they do not discuss the new WHO report that was published in 2019. CDC also published a very important and relevant work in 2019 as well. 

CDC Antimicrobial resistance report: https://www.cdc.gov/drugresistance/pdf/threats-report/2019-ar-threats-report-508.pdf 

WHO Antimicrobial resistant report: https://www.who.int/antimicrobial-resistance/interagency-coordination-group/IACG_final_report_EN.pdf?ua=1 

-List of references are not current and very few studies are included from 2019 and 2020.  

Author Response

There are several sentences throughout the manuscript that are identical to the existing literature and raises concern about overall syntax and writing of the review manuscript. There are also more than one font used throughout the manuscript. 

Re: Our plagiarism checker (part of our University tools, i.e. unicheck) identified 0.91% of coincidences of our manuscript with the literature. Those identified sentences were to reference a particular finding that we decided not to detail because space limitations. In any case, in the novel version we have rephrased as much as possible.

-Abstract of the study is not providing the results of comprehensive analysis and review of the literature and simply state the problem rather than discussing outcomes derived from this review study. 

Re: We have changed the abstract.

-There are major and important microbiological errors in the manuscript, for example, authors indicate Salmonella spp on line 29 that is not a correct nomenclature. This genus is now has only 2 species (bongri and enterica) and S. bongori is not a human pathogen so all serovars of Salmonella are derived from one species, hence Salmonella spp. is incorrect terminology and should be replaced by Salmonella serovars. This is a common mistake in the literature and CDC does a great job explaining this at: 

https://wwwnc.cdc.gov/eid/page/scientific-nomenclature  

Re: thank you. The reviewer is correct. We have considered his/her point.

-Key references are not discussed, although authors discuss a WHO report, they do not discuss the new WHO report that was published in 2019. CDC also published a very important and relevant work in 2019 as well. 

CDC Antimicrobial resistance report: https://www.cdc.gov/drugresistance/pdf/threats-report/2019-ar-threats-report-508.pdf 

WHO Antimicrobial resistant report: https://www.who.int/antimicrobial-resistance/interagency-coordination-group/IACG_final_report_EN.pdf?ua=1 

Re: We have introduced data related to the new WHO report, however note that we have varied the focus of the work based on comments of reviewer 3, and we do not only focus on MDR.

-List of references are not current and very few studies are included from 2019 and 2020.  

Re: We referenced 50 works of the last 2 years (17 published in 2018, 22 published in 2019, and 11 published in 2020) which, considering that we discussed different mechanisms (earlier described), is a substantial update in the field.

Reviewer 3 Report

This manuscript reviews antibiotic resistance and tolerance in biofilms wich is an area of interest. I have found some typos and I have some small comments that could improve the text. I would recommend a careful review of the text to identify any other that might have gone unnoticed.

Line 8 multi-drug resistant (MDR) bacteria

Line 10 antibiotic treatment

Line 11 antibiotic resistance genes

Line 18 MDR

Line 28 their

Line 32-33 and this number is expected to exceed 10 million deaths by 2050

Line 45 rephrase sentence about Neisseria gonorrhoeae

Line 63 adhered to a biotic or an abiotic surface

Line 74 result in

Line 90 why did the authors decide to focus on how MDR bacteria survive antibiotics in biofilms? Wouldn’t most of the text be also relevant for non MDR bacteria? ECM, heterogeneous physiology…

Line 103 referred to as

Line 105 coordinated

Line 107 from the cell surface

Line 111 helps

Line 117 end up

Line 118 become

Line 134 negatively

Line 141 whose

Line 150 for instance

Line 174 strengthening

Line 177 what do the authors mean by cognate substrates?

Line 187 keeps

Line 191 the ECD additionally retains water,

Line 194 , for instance,

Line 199 I don’t think intervenient is an appropriate word in this sentence

Line 204 remove ‘the’ from the figure 1

Line 206 ATCC 15692

Line 207 ATCC 51299

Line 215 surface exposed proteins

Line 216 whose

line 222 biofilm recalcitrance

line 224 grow

line 224 long periods of time and is quantifiable

line 227 once it is in the

Line 240 their function is no longer essential, the microorganism

Line 251 cannot grow

Line 242 figure 2, without the, at the beginning

Line 244 does not depend on one unique

Line 246 remove depend

Line 250 the efficacy of antibiotics in biofilm forming cells

Line 251 generates

The authors redundantly repeat many times throughout the manuscript that parameters are strain or bacteria dependent. This idea does not need to be repeated in every paragraph, this can be stated at the beginning or the end of the paper just enunciating that:

- the time for an antibiotic to reach the interior of the biofilm,

- The ECM biogenesis and composition

- line 316 AMR offered by ECM

Is strain or bacteria dependent

Line 267 against aminoglycosides

Line 268 unlike alginate

Line 277 enhances aminoglycoside resistance

Line 278 or β-lactams. eDNA also enhances

Line 286 decreasing the pH

Line 287 and S. Typhimurium, reduction

Line 287 trigger

Line 288 activate

Line 288 describe AMR

Line 300 promote

Line 304 it is tempting

Lien 305 occur

Line 310 alter

Line 311 thus,

Line 314 furthermore,

Line 318 conferred

Line 330 there are two types of persister bacteria, type I or triggered and type II or spontaneous persisters, the last one can divide during antibiotic treatment

Line 331 resume growth

Lime 331 antibiotic removal

Line 337 that target

Line 338 thus,

Line 338 slow growing

Line 339 acts on the membrane (regardless of whether it is compromised)

Line 340 respond

Line 342 amino acid, carbon and iron starvation

Figure 3 is pixeled, please export with higher quality

Figure 3 amino acid, carbon and iron deprivation

Line 350 shuts down almost all metabolic processes

Line 350 become

Line 351 has been linked

Line 354 has been related to

Line 354 related to killing or to susceptibility to these antibiotics?

Line 379 allows the toxins to exert

Line 38q several studies have snown

Line 384 enhances

Line 386 decreases

Line 396 the SOS response and SR participate

Line 402 the SOS response and the SR

Line 413 anaerobic conditions, microorganisms

Line 418 require

Line 425 oxidize

Line 420 I believe this should state by increasing cellular hydroxyl radical levels

Line 426 stimulate

Line 426 catalases detoxify H2O2 and SOD detoxifies superoxide anion. Does the fenton reaction also generate these ROS?

Line 429 represses

Line 434 more highly activated in

Line 435 biofilm cells

Line 446, 454 what do you mean by propellers?

Line 451 transports

Line 458 that, like SMR members, utilize

Line 459 facilitates

Line 460 and phosphate

Line 461 tetracyclines and fluoroquinolones

Line 465 transports

Line 467 share

Line 475 AcrAB-TolC

Line 482 of the LsrABCD complex

Line 483 the mdpF gene

Line 494 mediates

Line 501 contribute

Line 503 exposure

Line 509 alter

Line 518 makes cells resistant

Line 519 efflux pump production

Line 530 activate

Line 543 different substrate specificities

Line 546 studies

Line 548 participate

Linen 564 and wild type bacteria

Line 583 when they coinfect

Line 583 recognizes

Line 584 induces

Line 584 regulates

Line 596 functions

Line 599 where it displays toxic activities

Line 600 inhibits

Line 600 this has an important role

Line 608 what do the authors mean by contemporary bacteria?

Line 620 establishes

Line 623 increases

Line 627 within the biofilm environment

Line 634 apart from

Line 636 MDR-related bacteria

Line 641 that compared

Line 641 biofilm cells

Line 654 this sentence is a little bit confusing, I think it is better to rephrase it

Line 658 their relevance

Line 661 thus,

Line 686 has been reported

Line 689 may be

Line 701 to the best of our knowledge

Line 718 resistant

Line 721 contribute

Line 735 dispersers

Line 768 disperses

Line 770 other molecule

Line 770 destabilizes

Line 771 inhibits

Line 774 biofilms

Line 793 detergent-like

Line 804 their application

Line 814 their effectivity

Line 817 they are of small size

Line 829 supported by

Line 834 preventing them from reaching

Line 838 inactivate

Line 845 stimulates

Line 845 the emergence of phage resistant populations to… has been reported

Line 847 activities, only some

Line 855 comprise complex mixtures

Line 874 synergistic

Line 877 their extraction

Line 881 has led

Line 883 contributes

Line 886 heterogeneity

Line 890 studying them

Line 891 challenging

Line 894 mediate

Line 895 successfully deal

4- Light and other physical mechanisms have also been used to control biofilm infections. The authors might want to consider adding a small remark about them.

Author Response

This manuscript reviews antibiotic resistance and tolerance in biofilms wich is an area of interest. I have found some typos and I have some small comments that could improve the text. I would recommend a careful review of the text to identify any other that might have gone unnoticed.

Line 8 multi-drug resistant (MDR) bacteria
Line 10 antibiotic treatment
Line 11 antibiotic resistance genes
Line 18 MDR
Line 28 their
Line 32-33 and this number is expected to exceed 10 million deaths by 2050
Line 45 rephrase sentence about Neisseria gonorrhoeae
Line 63 adhered to a biotic or an abiotic surface
Line 74 result in
Line 90 why did the authors decide to focus on how MDR bacteria survive antibiotics in biofilms? Wouldn’t most of the text be also relevant for non MDR bacteria? ECM, heterogeneous physiology…
Line 103 referred to as
Line 105 coordinated
Line 107 from the cell surface
Line 111 helps
Line 117 end up
Line 118 become
Line 134 negatively
Line 141 whose
Line 150 for instance
Line 174 strengthening
Line 177 what do the authors mean by cognate substrates?
Line 187 keeps
Line 191 the ECD additionally retains water,
Line 194 , for instance,
Line 199 I don’t think intervenient is an appropriate word in this sentence
Line 204 remove ‘the’ from the figure 1
Line 206 ATCC 15692
Line 207 ATCC 51299
Line 215 surface exposed proteins
Line 216 whose
line 222 biofilm recalcitrance
line 224 grow
line 224 long periods of time and is quantifiable
line 227 once it is in the
Line 240 their function is no longer essential, the microorganism
Line 251 cannot grow
Line 242 figure 2, without the, at the beginning
Line 244 does not depend on one unique
Line 246 remove depend
Line 250 the efficacy of antibiotics in biofilm forming cells
Line 251 generates
The authors redundantly repeat many times throughout the manuscript that parameters are strain or bacteria dependent. This idea does not need to be repeated in every paragraph, this can be stated at the beginning or the end of the paper just enunciating that:
- the time for an antibiotic to reach the interior of the biofilm,
- The ECM biogenesis and composition
- line 316 AMR offered by ECM
Is strain or bacteria dependent
Line 267 against aminoglycosides
Line 268 unlike alginate
Line 277 enhances aminoglycoside resistance
Line 278 or β-lactams. eDNA also enhances
Line 286 decreasing the pH
Line 287 and S. Typhimurium, reduction
Line 287 trigger
Line 288 activate
Line 288 describe AMR
Line 300 promote
Line 304 it is tempting
Lien 305 occur
Line 310 alter
Line 311 thus,
Line 314 furthermore,
Line 318 conferred
Line 330 there are two types of persister bacteria, type I or triggered and type II or spontaneous persisters, the last one can divide during antibiotic treatment
Line 331 resume growth
Lime 331 antibiotic removal
Line 337 that target
Line 338 thus,
Line 338 slow growing
Line 339 acts on the membrane (regardless of whether it is compromised)
Line 340 respond
Line 342 amino acid, carbon and iron starvation
Figure 3 is pixeled, please export with higher quality
Figure 3 amino acid, carbon and iron deprivation
Line 350 shuts down almost all metabolic processes
Line 350 become
Line 351 has been linked
Line 354 has been related to
Line 354 related to killing or to susceptibility to these antibiotics?
Line 379 allows the toxins to exert
Line 38q several studies have snown
Line 384 enhances
Line 386 decreases
Line 396 the SOS response and SR participate
Line 402 the SOS response and the SR
Line 413 anaerobic conditions, microorganisms
Line 418 require
Line 425 oxidize
Line 420 I believe this should state by increasing cellular hydroxyl radical levels
Line 426 stimulate
Line 426 catalases detoxify H2O2 and SOD detoxifies superoxide anion. Does the fenton reaction also generate these ROS?
Line 429 represses
Line 434 more highly activated in
Line 435 biofilm cells
Line 446, 454 what do you mean by propellers?
Line 451 transports
Line 458 that, like SMR members, utilize
Line 459 facilitates
Line 460 and phosphate
Line 461 tetracyclines and fluoroquinolones
Line 465 transports
Line 467 share
Line 475 AcrAB-TolC
Line 482 of the LsrABCD complex
Line 483 the mdpF gene
Line 494 mediates
Line 501 contribute
Line 503 exposure
Line 509 alter
Line 518 makes cells resistant
Line 519 efflux pump production
Line 530 activate
Line 543 different substrate specificities
Line 546 studies
Line 548 participate
Linen 564 and wild type bacteria
Line 583 when they coinfect
Line 583 recognizes
Line 584 induces
Line 584 regulates
Line 596 functions
Line 599 where it displays toxic activities
Line 600 inhibits
Line 600 this has an important role
Line 608 what do the authors mean by contemporary bacteria?
Line 620 establishes
Line 623 increases
Line 627 within the biofilm environment
Line 634 apart from
Line 636 MDR-related bacteria
Line 641 that compared
Line 641 biofilm cells
Line 654 this sentence is a little bit confusing, I think it is better to rephrase it
Line 658 their relevance
Line 661 thus,
Line 686 has been reported
Line 689 may be
Line 701 to the best of our knowledge
Line 718 resistant
Line 721 contribute
Line 735 dispersers
Line 768 disperses
Line 770 other molecule
Line 770 destabilizes
Line 771 inhibits
Line 774 biofilms
Line 793 detergent-like
Line 804 their application
Line 814 their effectivity
Line 817 they are of small size
Line 829 supported by
Line 834 preventing them from reaching
Line 838 inactivate
Line 845 stimulates
Line 845 the emergence of phage resistant populations to… has been reported
Line 847 activities, only some
Line 855 comprise complex mixtures
Line 874 synergistic
Line 877 their extraction
Line 881 has led
Line 883 contributes
Line 886 heterogeneity
Line 890 studying them
Line 891 challenging
Line 894 mediate
Line 895 successfully deal

Re: thank you so much for point out all these typos. We have considered all and review the entire manuscript.
Regarding the point on MDR, the reviewer is correct. Focused on MDR may affect the  understanding of the reader suggesting that the discussed biofilm mechanisms only apply to MDR bacteria. Thus, in the novel version of the manuscript we have changed this focus. We have modified the article title, removed the previous table and changed the focus of the work  through the entire manuscript.

4- Light and other physical mechanisms have also been used to control biofilm infections. The authors might want to consider adding a small remark about them.
Re: We have included a section dedicated to this topic.

Round 2

Reviewer 2 Report

This manuscript is now revised to great extent. Previously raised issues including similarity index are now addressed, the new article has SI of below 9% (excluding the references). New references are added, and previous comments were incorporated. Considering that the manuscript is now improved drastically across all sections, it could be further considered for publication.